# Improved Convergence in High Probability of Clipped Gradient Methods with Heavy Tailed Noise

**Ta Duy Nguyen**
Department of Computer Science
Boston University
taduy@bu.edu

**Thien Hang Nguyen**
Khoury College of Computer Sciences
Northeastern University
nguyen.thien@northeastern.edu

**Alina Ene**
Department of Computer Science
Boston University
aene@bu.edu

**Huy Le Nguyen**
Khoury College of Computer Sciences
Northeastern University
hu.nguyen@northeastern.edu

## Abstract

In this work, we study the convergence *in high probability* of clipped gradient methods when the noise distribution has heavy tails, i.e., with bounded $p$th moments, for some $1 < p \leq 2$. Prior works in this setting follow the same recipe of using concentration inequalities and an inductive argument with union bound to bound the iterates across all iterations. This method results in an increase in the failure probability by a factor of $T$, where $T$ is the number of iterations. We instead propose a new analysis approach based on bounding the moment generating function of a well chosen supermartingale sequence. We improve the dependency on $T$ in the convergence guarantee for a wide range of algorithms with clipped gradients, including stochastic (accelerated) mirror descent for convex objectives and stochastic gradient descent for nonconvex objectives. Our high probability bounds achieve the optimal convergence rates and match the best currently known in-expectation bounds. Our approach naturally allows the algorithms to use time-varying step sizes and clipping parameters when the time horizon is unknown, which appears difficult or even impossible using existing techniques from prior works. Furthermore, we show that in the case of clipped stochastic mirror descent, several problem constants, including the initial distance to the optimum, are not required when setting step sizes and clipping parameters.

## 1 Introduction

Stochastic optimization is a well-studied area with many applications ranging from machine learning, to operation research, numerical linear algebra and beyond. In contrast to deterministic algorithms, stochastic algorithms might fail, and a pertinent question is how often does failure happen and how to increase the success rate. These questions are especially important in critical applications where failure is not tolerable, or when a single run is costly in time and resources. Fortunately, the standard stochastic gradient descent (SGD) algorithm has been shown to converge with *high probability* under a *light-tailed noise* distribution such as sub-Gaussian distributions [23, 12, 27, 14, 11, 10, 18], which gives strong guarantee on the success of single runs. However, recent observations in popular deep learning applications, such as training attention models [33] and convolutional networks [30], reveal a more challenging optimization landscape: the gradient noises follow *heavy-tailed* distributions, where the variance may be infinite [29, 33, 9], whereas

37th Conference on Neural Information Processing Systems (NeurIPS 2023).

the standard light-tailed setting assumes that all the moments are bounded. Heavy-tailed gradient noises can cause algorithms like SGD to fail, and this mismatch between theory and practice has been suggested to be one of the reasons for the strong preference of adaptive methods like Adam over SGD in modern settings [33].

In this work, we consider the setting of *heavy-tailed noise* proposed by Zhang et al., (2020) [33], where the (unbiased) gradient noise only has bounded $p$th moments, for some $p \in (1, 2]$. While standard SGD can fail to converge when the variance is unbounded, i.e. when $p < 2$, [33] show that SGD with appropriate clipping (or *Clipped-SGD*) converges *in expectation* under heavy-tailed noise, where the convergence rate depends on $O\left(\frac{1}{\delta}\right)$ if $\delta$ is the targeted maximum failure probability. It is more desirable, however, to obtain convergence results in *high probability,* where the convergence rate depends instead on $O(\log \frac{1}{\delta})$, which gives better guarantees for single runs.

Recent follow-up works [2, 28, 19] show that variants of Clipped-SGD in fact converge with high probability. This is a pleasing result, extending the earlier work by [7] for $p = 2$. However, there are several shortcomings of these results when compared with the corresponding bounds in the light-tailed setting. First, the clipped algorithm uses a fixed step size and a fixed clipping parameter depending on the number of iterations, which precludes results with *unknown* time horizons. Secondly, the convergence guarantees are worse than the light-tailed bounds by a $\log T$ factor, even for fixed step sizes and clipping parameters. These issues beg a qualitative question:

*Is heavy-tailed noise inherently harder than light-tailed noise?*

In this work, we answer the above question for Clipped-SGD and the general clipped (accelerated) stochastic mirror descent (*Clipped-SMD*) algorithm. We give an improved analysis framework that not only gives tighter bounds matching the light-tailed noise setting, but also allows for step sizes and clipping parameters for unknown time horizons. Furthermore, we show that this framework is applicable to various settings, from finding minimizers of convex functions with arbitrarily large domains using (accelerated) mirror descent, to finding stationary points for non-convex functions using gradient descent.

## 1.1 Contributions and Techniques

Our work addresses several open questions posed by previous works including handling general domains and dealing with an unknown time horizon under heavy-tailed noise. Qualitatively, we close the logarithmic suboptimality gap and achieve the optimal rate in several settings. More specifically:

− We demonstrate a novel approach to analyze clipped gradient methods in high probability that is general and applies to various standard settings. In the convex setting, we analyze Clipped-SMD and clipped stochastic accelerated mirror descent. In the non-convex setting, we analyze Clipped-SGD. Using our new analysis, we show that clipped methods attain time-optimal convergence in high probability for both convex and nonconvex objectives under heavy-tailed gradient noise. In the convex setting, we obtain an $O\left(T^{\frac{1-p}{p}}\right)$ convergence rate for arbitrary (not necessarily compact) convex domains for Clipped-SMD and $O\left(T^{\frac{1-p}{p}}\sigma + T^{-2}\right)$ for accelerated Clipped-SMD, where $\sigma$ is the noise parameter. These rates are time-optimal and match the lower bounds in [26, 31]. In the nonconvex setting, we obtain the optimal convergence rate of $O\left(T^{\frac{2-2p}{3p-2}}\right)$ for clipped-SGD. This bound is also time-optimal and matches the lower bound in [33]; it also complements the in-expectation convergence of clipped-SGD provided by [33].

− Previous works for heavy-tailed noises follow the recipe of using Freedman-type inequalities [4, 3] as a *blackbox* and bound the iterates inductively for all iterations. This process incurs an additional $\log T$ dependency in the final convergence rate; in other words, the success probability goes from $1 - \delta$ to $1 - T\delta$. The step sizes and clipping parameters of this approach depend on the time horizon $T$ to enable the union bound and induction across all iterations in the analysis, excluding the important case when the time horizon is unknown. Our whitebox approach forgoes the aforementioned induction, not only circumventing the $\log T$ loss but also allowing for an unknown time horizon. We further show that our analysis allows for a choice of step size and clipping parameters that do not depend on generally unknown parameters like the noise-parameter $\sigma$, the

failure probability $\delta$, and the initial distance to the optimum, all of which appear impossible using only the techniques from prior works.

− Our whitebox approach analyzes the moment generating function of a well chosen martingale difference sequence to obtain tight rates for stochastic gradient methods. This approach is closest to the work of [18], which only work in the light-tailed noise setting. In contrast to the light-tailed noise setting where all the moments are well controlled, the heavy-tailed setting often requires algorithms to incorporate gradient clipping for controlling the possibly infinite moments. However, this makes the gradient estimate biased and requires more careful attention to control the bias propagating through the algorithm. Naively applying the technique in [18] is not enough to handle heavy-tailed noise. Rather, as will be shown in our analysis, we introduce a novel history-dependent weights for the martingale sequence that is able to cope with the propagating bias term of clipped methods for heavy-tailed noise across various settings.

## 1.2 Related Works

**High probability convergence for light-tailed noises.** Convergence in high probability of stochastic gradient algorithms has been established for sub-Gaussian noises in a number of prior works, including [23, 12, 27, 14, 11, 10] for convex problems with bounded domain (or bounded Bregman diameter) or with strong convexity. Other works [17, 20, 16] study convergence of variants of SGD for nonconvex objectives, where they consider sub-Gaussian and sub-Weibull noises. The most relevant to ours in this line of work is the one by [18], where a whitebox approach is employed to obtain tight rates for stochastic gradient methods in the light-tailed noise setting. However, their technique is not directly applicable in the heavy-tailed noise setting, where we need to introduce new ideas to handle the biases introduced by gradient clipping.

**High probability convergence for noises with bounded variance and heavy tails.** The design of new gradient algorithms and their analysis in the presence of heavy-tailed noises has drawn significant recent interest. Starting from the work [25] which propose Clipped-SGD to handle exploding gradients in recurrent neural networks, the recent works [30, 29, 33, 9] give new motivation for clipped methods in the context of convolutional networks and attention deep networks that attempts to explain the dominance of adaptive methods over SGD in practical modern scenarios.

While the convergence in expectation of vanilla SGD has been extensively studied [5, 23, 13, 18], only recently has the convergence of Clipped-SGD with heavy tailed noises been closely examined. There, [33] first show the convergence in expectation of Clipped-SGD for nonconvex functions and provide a matching lower bound. In the convex regime, several works with different clipping strategies for the case of $p = 2$ have shown high probability convergence for smooth problems with bounded domain [22, 24], smooth unconstrained problems [7], and non-smooth problems [8]. A variant of Clipped-SGD that utilizes momentum [2] has also been shown to converge with high probability for bounded $p$th moments gradient noise. However, the analysis in [2] requires a strong assumption which implies that the true gradients are bounded, a restrictive assumption that excludes objectives like quadratic functions.

More recently, [28, 19, 34] give nearly-optimal convergence rates for several Clipped-SGD variants. These works follow the recipe of using Freedman-type inequalities [4, 3] as a blackbox and bound the iterates inductively for all iterations, which incur an additional $\log T$ dependency in the final convergence rate. We show in our work that existing convergence rates can be tightened up and improved. Tight lower bounds for the optimal convergence rate have been shown by [26, 31] for convex objectives and by [33] for nonconvex settings. In both cases, our paper provides optimal convergence guarantees.

In a related but different line of work, [32] show that vanilla SGD can converge with heavy tailed noise for a special type of strongly convex functions, and [31] show that stochastic mirror descent converges in expectation for a special choice of mirror maps, although only for strongly convex objectives with bounded domains.

## 2 Preliminaries: Assumptions and Notations

We study the problem $\min_{x \in \mathcal{X}} f(x)$ where $f : \mathbb{R}^d \to \mathbb{R}$ and $\mathcal{X}$ is the domain of the problem. In the convex setting, we assume that $\mathcal{X}$ is a convex set but not necessarily compact. We let $\|\cdot\|$ be an

Table 1: Previous and new results for high-probability convergence (with failure probability $\delta$) of clipped SMD and SGD under heavy tailed noise: $\mathbb{E}[\|\widehat{\nabla}f(x)-\nabla f(x)\|_*^p \mid x] \leq \sigma^p$ for some $p \in (1,2]$, where $\widehat{\nabla}f(x)$ denotes the stochastic gradient at $x$ for the objective $f$. For the convex setting, the error bounds are for the optimality gaps $\frac{1}{T}\sum_{t=1}^T f(x_t) - f^*$. For the nonconvex setting, we bound the gradient norm $\frac{1}{T}\sum_{t=1}^T \|\nabla f(x_t)\|^2$. Here, $\widetilde{O}(\cdot)$ hides polylog $T$ factors. Note that, for simplicity, we do not compare against results in more specialized settings such as bounded domain or bounded gradients, as well as other variants of clipped SGD.

| | Assumptions | Convex Setting (Clipped-SMD) | Non-convex Setting (Clipped-SGD) |
|---|---|---|---|
| Lower bound | $p \in (1,2]$ | $\Omega\left(T^{\frac{1-p}{p}}\right)$ [31] | $\Omega\left(T^{\frac{-2(p-1)}{3p-2}}\right)$ [33] |
| Previous high-probability results | Known $T$ | $\widetilde{O}\left(T^{\frac{1-p}{p}}\right)$ [28] | $\widetilde{O}\left(T^{\frac{1-p}{p}}\right)$ [28] |
| Our results | Known $T$ | $O\left(T^{\frac{1-p}{p}}\right)$ (Thm 4.1) | $O\left(T^{\frac{2-2p}{3p-2}}\right)$ (Thm 3.1) |
| | Unknown $T$ | $\widetilde{O}\left(T^{\frac{1-p}{p}}\right)$ (Thm 4.4) | $\widetilde{O}\left(T^{\frac{2-2p}{3p-2}}\right)$ (Thm B.2) |

arbitrary norm and $\|\cdot\|_*$ be its dual norm. In the nonconvex setting, we take $\mathcal{X}$ to be $\mathbb{R}^d$ and consider only the $\ell_2$ norm.

## 2.1 Assumptions

Our paper works with the following assumptions:

**(1) Existence of a minimizer**: In the convex setting, we assume that there exists $x^* \in \arg\min_{x \in \mathcal{X}} f(x)$. We let $f^* = f(x^*)$.

**(1') Existence of a finite lower bound**: In the nonconvex setting, we assume that $f$ admits a finite lower bound, i.e., $f^* := \inf_{x \in \mathbb{R}^d} f(x) > -\infty$.

**(2) Unbiased estimator**: We assume that our algorithm is allowed to query a stochastic first-order oracle that returns a history-independent, unbiased gradient estimator $\widehat{\nabla}f(x)$ of $\nabla f(x)$ for any $x \in \mathcal{X}$. That is, conditioned on the history and the queried point $x$, we have $\mathbb{E}[\widehat{\nabla}f(x) \mid x] = \nabla f(x)$.

**(3) Bounded $p$th moment noise**: We assume that there exists $\sigma > 0$ such that for some $1 < p \leq 2$ and for any $x \in \mathcal{X}$, $\widehat{\nabla}f(x)$ satisfies $\mathbb{E}[\|\widehat{\nabla}f(x) - \nabla f(x)\|_*^p \mid x] \leq \sigma^p$.

**(4) $L$-smoothness**: We consider the class of $L$-smooth functions: for all $x, y \in \mathbb{R}^d$, $\|\nabla f(x) - \nabla f(y)\|_* \leq L \|x - y\|$.

## 2.2 Gradient Clipping Operator and Notations

We introduce the gradient clipping operator and its general properties used in Clipped-SMD (Algorithm 2) and Clipped-SGD (Algorithm 1). Let $x_t$ be the output at iteration $t$ of an algorithm of interest. We denote by $\widehat{\nabla}f(x_t)$ the stochastic gradient obtained by querying the gradient oracle. The clipped gradient estimate $\widetilde{\nabla}f(x_t)$ is taken as

$$\widetilde{\nabla}f(x_t) = \min\left\{1, \frac{\lambda_t}{\left\|\widehat{\nabla}f(x_t)\right\|_*}\right\}\widehat{\nabla}f(x_t), \tag{1}$$

where $\lambda_t$ is the clipping parameter used in iteration $t$. In subsequent sections, we let $\Delta_t := f(x_t) - f^*$ denote the optimal function value gap at $x_t$. We let $\mathcal{F}_t = \sigma\left(\widehat{\nabla}f(x_1), \ldots, \widehat{\nabla}f(x_t)\right)$ be the natural filtration at time $t$ and define the following notations for the stochastic error, the deviation, and the bias of the clipped gradient estimate at time $t$:

$$\theta_t = \widetilde{\nabla}f(x_t) - \nabla f(x_t); \quad \theta_t^u = \widetilde{\nabla}f(x_t) - \mathbb{E}\left[\widetilde{\nabla}f(x_t) \mid \mathcal{F}_{t-1}\right]; \quad \theta_t^b = \mathbb{E}\left[\widetilde{\nabla}f(x_t) \mid \mathcal{F}_{t-1}\right] - \nabla f(x_t).$$

Note that $\theta_t^u + \theta_t^b = \theta_t$. Regardless of the convexity of the function $f$, the following lemma provides upper bounds for these quantities. These bounds can be found in prior works [7, 33, 19, 28] for the

**Algorithm 1** Clipped-SGD

---

Parameters: initial point $x_1$, step sizes $\{\eta_t\}$, clipping parameters $\{\lambda_t\}$
for $t = 1$ to $T$ do

$$\widetilde{\nabla} f(x_t) = \min \left\{ 1, \frac{\lambda_t}{\|\widehat{\nabla} f(x_t)\|} \right\} \widehat{\nabla} f(x_t)$$

$$x_{t+1} = x_t - \eta_t \widetilde{\nabla} f(x_t)$$

---

special case of $\ell_2$ norm. The extension to the general norm follows in the same manner, which we omit in this work.

**Lemma 2.1.** *For stochastic gradients $\widehat{\nabla} f(x_t)$ with bounded pth moment noise, the clipped gradients $\widetilde{\nabla} f(x_t)$ satisfy the following properties:*

$$\|\theta_t^u\|_* = \left\| \widetilde{\nabla} f(x_t) - \mathbb{E}\left[ \widetilde{\nabla} f(x_t) \mid \mathcal{F}_{t-1} \right] \right\|_* \leq 2\lambda_t. \tag{2}$$

*Furthermore, if $\|\nabla f(x_t)\|_* \leq \frac{\lambda_t}{2}$ then*

$$\left\| \theta_t^b \right\|_* = \left\| \mathbb{E}\left[ \widetilde{\nabla} f(x_t) \mid \mathcal{F}_{t-1} \right] - \nabla f(x_t) \right\|_* \leq 4\sigma^p \lambda_t^{1-p}; \tag{3}$$

$$\mathbb{E}\left[ \|\theta_t^u\|_*^2 \right] = \mathbb{E}\left[ \left\| \widetilde{\nabla} f(x_t) - \mathbb{E}_t\left[ \widetilde{\nabla} f(x_t) \right] \right\|_*^2 \mid \mathcal{F}_{t-1} \right] \leq 40\sigma^p \lambda_t^{2-p}. \tag{4}$$

Finally, we state a simple but important lemma that bounds the moment generating function of a zero-mean bounded random variable. The proof can be found in, for example, equation (3) of [1].

**Lemma 2.2.** *Let $X$ be a random variable such that $\mathbb{E}[X] = 0$ and $|X| \leq R$ almost surely. Then for $0 \leq \lambda \leq \frac{1}{R}$*

$$\mathbb{E}\left[ \exp(\lambda X) \right] \leq \exp\left( \frac{3}{4} \lambda^2 \mathbb{E}\left[ X^2 \right] \right).$$

## 3 Clipped Stochastic Gradient Descent for Nonconvex Functions

In this section, we study the convergence of Clipped-SGD for nonconvex functions. Here, we consider the domain to be $\mathbb{R}^d$ equipped with the standard $\ell_2$ norm. We first outline a blackbox concentration argument to show convergence in high probability of Algorithm 1 and then follow-up with a more powerful whitebox approach that allows for a tight high probability convergence analysis.

**Comparison to previous works.** In the simple setting of known time horizon and without momentum for Clipped-SGD, the $\widetilde{O}(T^{\frac{2-2p}{3p-2}})$ convergence rate has not been shown before to the best of our knowledge. The recent work by [28] study this case and only give a suboptimal rate of $\widetilde{O}(T^{\frac{1-p}{p}})$. Note that [2, 19] study other variants of Clipped-SGD with momentums incorporated. Although [2, 19] achieve the nearly-optimal time dependency of $\widetilde{O}(T^{\frac{2-2p}{3p-2}})$ in the non-convex settings, they rely on using blackbox concentration inequalities which result in a suboptimal convergence rate that also requires a known time horizon.

We first present the guarantee for known time horizon $T$ via our whitebox approach in Theorem 3.1 and defer the statement for unknown $T$ in Theorem B.2 to the Appendix.

**Theorem 3.1.** *Assume that $f$ satisfies Assumption (1'), (2), (3), (4). Let $\gamma := \max \left\{ \log \frac{1}{\delta}; 1 \right\}$ and $\Delta_1 := f(x_1) - f^*$. For known time horizon $T$, we choose $\lambda_t$ and $\eta_t$ such that*

$$\lambda_t := \lambda := \max \left\{ \left( \frac{8\gamma}{\sqrt{L\Delta_1}} \right)^{\frac{1}{p-1}} T^{\frac{1}{3p-2}} \sigma^{\frac{p}{p-1}}; 2\sqrt{90L\Delta_1}; 32^{\frac{1}{p}} \sigma T^{\frac{1}{3p-2}} \right\}$$

$$\eta_t := \eta := \frac{\sqrt{\Delta_1} T^{\frac{1-p}{3p-2}}}{8\lambda\sqrt{L}\gamma} = \frac{\sqrt{\Delta_1}}{8\sqrt{L}\gamma} \min \left\{ \left( \frac{8\gamma}{\sqrt{L\Delta_1}} \right)^{\frac{-1}{p-1}} T^{\frac{-p}{3p-2}} \sigma^{\frac{-p}{p-1}}; \frac{T^{\frac{1-p}{3p-2}}}{2\sqrt{90L\Delta_1}}; \frac{T^{\frac{-p}{3p-2}}}{32^{1/p}\sigma} \right\}.$$

*Then with probability at least $1 - \delta$*

$$\frac{1}{T}\sum_{t=1}^{T}\|\nabla f(x_t)\|^2 \le 720\sqrt{\Delta_1 L}\gamma \max\left\{\left(\frac{8\gamma}{\sqrt{L\Delta_1}}\right)^{\frac{1}{p-1}}T^{\frac{2-2p}{3p-2}}\sigma^{\frac{p}{p-1}};\right.$$

$$\left.2\sqrt{90L\Delta_1}T^{\frac{1-2p}{3p-2}};32^{1/p}\sigma T^{\frac{2-2p}{3p-2}}\right\} = O\left(T^{\frac{2-2p}{3p-2}}\right).$$

*Remark* 3.2. In comparison to the corresponding results in [28] (Theorem E.2), while our result achieves a poly $T$ factor better rate when $p < 2$, the dependency on $\log\frac{1}{\delta}$ in our result contains a dependency on $p$ while the result in [28] does not. That term can dominate the convergence rate in the regime when $\delta$ is very small and $p$ is very close to 1. Hence, an open question is to remove such dependency on $p$ for the $\log\frac{1}{\delta}$ term while still maintain the optimal rate on $T$.

Now, we turn to the analysis, starting with the key Lemma 3.3 (proof in the Appendix).

**Lemma 3.3.** *Assume that $f$ satisfies Assumption (1'), (2), (3), (4) and $\eta_t \le \frac{1}{L}$ then for all $t \ge 1$,*

$$\frac{\eta_t}{2}\|\nabla f(x_t)\|^2 \le \Delta_t - \Delta_{t+1} + \left(L\eta_t^2 - \eta_t\right)\langle\nabla f(x_t), \theta_t^u\rangle + \frac{3\eta_t}{2}\left\|\theta_t^b\right\|^2$$

$$+ L\eta_t^2\left(\|\theta_t^u\|^2 - \mathbb{E}\left[\|\theta_t^u\|^2 \mid \mathcal{F}_{t-1}\right]\right) + L\eta_t^2\mathbb{E}\left[\|\theta_t^u\|^2 \mid \mathcal{F}_{t-1}\right]. \quad (5)$$

*Remark* 3.4. In Lemma 3.3, we decompose the RHS into appropriate terms that allow us to define a martingale. This lemma helps us understand why we can achieve a better convergence rate $O(T^{\frac{2-2p}{3p-2}})$ (for minimizing the norm squared of the gradient) in comparison to the best rate of $O(T^{\frac{1-p}{p}})$ in the convex setting. We focus on the error term $\langle\nabla f(x_t), \theta_t\rangle = \langle\nabla f(x_t), \theta_t^u\rangle + \langle\nabla f(x_t), \theta_t^b\rangle$ on the RHS of (5). Since this error contains the gradient $\nabla f(x_t)$, we leverage some of the gain $\|\nabla f(x_t)\|^2$ on the LHS of 5: we use Cauchy-Schwarz to bound $\langle\nabla f(x_t), \theta_t^b\rangle \le \frac{1}{2}\|\nabla f(x_t)\|^2 + \frac{1}{2}\|\theta_t^b\|^2$ and use the some of the gain to absorb the first term. Then setting our parameters $\lambda_t, \eta_t$ appropriately to balance the remaining terms helps us achieve the $O(T^{\frac{2-2p}{3p-2}})$ rate. Contrast this to the convex setting in the next section: the mismatch between the error term that contains the distance term $\|x^* - x_t\|$ and the gain term that contains the function value gap $f(x_t) - f^*$ prevents us from using the gain to absorb some of the error. Thus, this explains the convergence rate discrepancy between the convex case and the non-convex setting (see also Remark 4.6).

Before giving a sketch of our whitebox approach, we present a sketch of a blackbox argument that gives a nearly time-optimal convergence rate. This approach has an additional $\log T$ factor in the final rate but will serve as a point of comparison for our new techniques, which will close the logarithmic gap.

**Blackbox approach.** The key lies in the following lemma, which yields the near optimal $\widetilde{O}(T^{\frac{2-2p}{3p-2}})$ convergence rate of Clipped-SGD. In this case, we assume that the clipping parameters $\lambda_t$ and the step sizes $\eta_t$ are fixed. Note that the success probability is only $1 - T\delta$. This result uses Lemma 3.3 and Freedman's inequality (Theorem A.1) primarily as a *blackbox* to bound the error terms inductively by the initial function value gap to optimality.

**Lemma 3.5.** *For $1 \le N \le T + 1$, let $\eta_t = \eta$, $\lambda_t = \lambda$ (the specific choices are omitted here for brevity) and $E_N$ be the event that for all $k = 1, \ldots N$,*

$$L\eta^2\sum_{t=1}^{k-1}\|\theta_t^u\|^2 + \left(L\eta^2 - \eta\right)\sum_{t=1}^{k-1}\langle\nabla f(x_t), \theta_t^u\rangle + \frac{3\eta}{2}\sum_{t=1}^{k-1}\left\|\theta_t^b\right\|^2 \le \Delta_1.$$

*Then $E_N$ happens with probability at least $1 - \frac{(N-1)\delta}{T}$ for each $N \in [T + 1]$.*

With the above lemma, we can obtain a near-optimal convergence rate. However, this rate is still suboptimal due to the use of $T$ union bounds as part of the induction proof. We now discuss an improved analysis that closes the remaining gap.

**Whitebox approach.** Our whitebox approach defines a novel supermartingale difference sequence $Z_t$ (shown below) and analyzes its moment generating function from first principles. The sequence is

designed to leverage the structure of the problem and Clipped-SGD via carefully chosen decreasing weights $z_t$ (shown below).

$$Z_t := z_t \left( \frac{\eta_t}{2} \|\nabla f(x_t)\|^2 + \Delta_{t+1} - \Delta_t - \frac{3\eta_t}{2} \|\theta_t^b\|^2 - L\eta_t^2 \mathbb{E}\left[ \|\theta_t^u\|^2 \mid \mathcal{F}_{t-1} \right] \right)$$

$$- \left( 3z_t^2 L\eta_t^2 \Delta_t + 6L^2 z_t^2 \eta_t^4 \lambda_t^2 \right) \mathbb{E}\left[ \|\theta_t^u\|^2 \mid \mathcal{F}_{t-1} \right]$$

$$\text{where } z_t := \frac{1}{2P_t \eta_t \lambda_t \max_{i \leq t} \sqrt{2L\Delta_i} + 8Q_t L\eta_t^2 \lambda_t^2}$$

for $P_t, Q_t \in \mathcal{F}_{t-1} \geq 1$. We also define $S_t := \sum_{i=1}^{t} Z_i$. Note that by selecting $P_t, Q_t, \eta_t, \lambda_t$ appropriately so that $P_t \eta_t \lambda_t$ and $Q_t \eta_t^2 \lambda_t^2$ are constants (see for example the proof of Proposition 3.7 in the Appendix), we can ensure that the sequence $z_t$ is decreasing.

We now present Lemma 3.6 which is the main result for controlling the above martingale, whose proof will offer insights into the main technique in this paper. The technique to prove Lemma 3.6 is similar to the standard way of bounding the moment generating function in proving concentration inequalities, such as Freedman's inequality [4, 3]. The main challenge here is to find a way to leverage the structure of Clipped-SGD and choose the suitable coefficients $z_t$. Similarly to [18] where the authors analyze SGD with sub-Gaussian noise, we analyze the martingale difference sequence in a "whitebox" manner. In [18], however, thanks to the light-tailed noise, the weights $z_t$ can be chosen depending only on the problem parameters and independently of the algorithm history. On the other hand, to use Lemma 2.2, we have to make sure that $z_t \leq \frac{1}{R}$, where $R$ is an upper bound for the martingale elements. The key here is to choose $z_t$ depending on the past iterates, and use the function value gaps $\Delta_t$ to absorb the error incurred during the analysis. We give a proof sketch and defer the full version to the Appendix.

**Lemma 3.6.** *For any $\delta > 0$, let $E(\delta)$ be the event that for all $1 \leq k \leq T$*

$$\frac{1}{2} \sum_{t=1}^{k} z_t \eta_t \|\nabla f(x_t)\|^2 + z_k \Delta_{k+1} \leq z_1 \Delta_1 + \log \frac{1}{\delta} + \sum_{t=1}^{k} \frac{3z_t \eta_t}{2} \|\theta_t^b\|^2$$

$$+ \sum_{t=1}^{k} \left( (3z_t^2 L\eta_t^2 \Delta_t + 6L^2 z_t^2 \eta_t^4 \lambda_t^2 + z_t L\eta_t^2) \mathbb{E}\left[ \|\theta_t^u\|^2 \mid \mathcal{F}_{t-1} \right] \right).$$

*Then $\Pr[E(\delta)] \geq 1 - \delta$.*

*Proof Sketch.* Using Lemmas 3.3, 2.2, and the condition for $z_t$, we can show that $\mathbb{E}\left[ \exp(Z_t) \mid \mathcal{F}_{t-1} \right] \leq 1$. This then implies

$$\mathbb{E}\left[ \exp(S_t) \mid \mathcal{F}_{t-1} \right] = \exp(S_{t-1}) \mathbb{E}\left[ \exp(Z_t) \mid \mathcal{F}_{t-1} \right] \leq \exp(S_{t-1}),$$

which means $(\exp(S_t))_{t \geq 1}$ is a supermartingale. By Ville's inequality, we have, for all $k \geq 1$, $\Pr\left[ S_k \geq \log \frac{1}{\delta} \right] \leq \delta \mathbb{E}\left[ \exp(S_1) \right] \leq \delta$. In other words, with probability at least $1 - \delta$, for all $k \geq 1$, $\sum_{t=1}^{k} Z_t \leq \log \frac{1}{\delta}$. Plugging in the definition of $Z_t$ we obtain the desired inequality. $\qquad\square$

We now specify the choice of $\eta_t$ and $\lambda_t$. The following lemma gives a general condition for the choice of $\eta_t$ and $\lambda_t$ that gives the right convergence rate in time $T$.

**Proposition 3.7.** *We assume that the event $E(\delta)$ from Lemma 3.6 happens. Suppose that for some $\ell \leq T$, there are constants $C_1, C_2$ and $C_3$ such that for all $t \leq \ell$*

*1. $\lambda_t \eta_t \sqrt{2L} \leq C_1$;   2. $\frac{1}{L\eta_t} \left( \frac{1}{\lambda_t} \right)^p \leq C_2$;   3. $\sum_{t=1}^{T} L \left( \frac{1}{\lambda_t} \right)^p \lambda_t^2 \eta_t^2 \leq C_3$;   4. $\|\nabla f(x_t)\| \leq \frac{\lambda_t}{2}$.*

*Then for all $t \leq \ell + 1$*

$$\frac{1}{2} \sum_{i=1}^{t} \eta_i \|\nabla f(x_i)\|^2 + \Delta_{t+1} \leq \left( \sqrt{\Delta_1} + 2\sqrt{A}C_1 \right)^2$$

*for a constant $A \geq \max \left\{ 64 \left( \log \frac{1}{\delta} + \frac{60\sigma^p C_3}{C_1^2} \right)^2 + \frac{48\sigma^{2p} C_2 C_3 + 140\sigma^p C_3}{C_1^2}; 1 \right\}$.*

Finally, the proof for Theorem 3.1 is a direct consequence of Proposition 3.7 where we defer the details to the Appendix.

**Algorithm 2** Clipped-SMD
___
Parameters: initial point $x_1$, step sizes $\{\eta_t\}$, clipping parameters $\{\lambda_t\}$, $\psi$ is 1-strongly convex wrt $\|\cdot\|$

for $t = 1$ to $T$ do

$$\widetilde{\nabla} f(x_t) = \min\left\{1, \frac{\lambda_t}{\|\widehat{\nabla} f(x_t)\|_*}\right\} \widehat{\nabla} f(x_t)$$

$$x_{t+1} = \arg\min_{x \in \mathcal{X}} \left\{\eta_t \left\langle \widetilde{\nabla} f(x_t), x \right\rangle + \mathbf{D}_\psi(x, x_t)\right\}$$
___

# 4 Clipped Stochastic Mirror Descent for Convex Objectives

In this section, we present and analyze the Clipped Stochastic Mirror Descent algorithm (Algorithm 2) under heavy-tailed noise, with a general domain and arbitrary norm.

We define the Bregman divergence $\mathbf{D}_\psi(x, y) = \psi(x) - \psi(y) - \langle \nabla \psi(y), x - y \rangle$, where $\psi : \mathbb{R}^d \to \mathbb{R}$ is a 1-strongly convex differentiable function with respect to the norm $\|\cdot\|$ on $\mathcal{X}$. We assume for convenience that $\mathrm{dom}(\psi) = \mathbb{R}^d$. Algorithm 2 is a generalization of Clipped-SGD for convex functions to an arbitrary norm. The only difference from the standard Stochastic Mirror Descent algorithm is the use of the clipped gradient $\widetilde{\nabla} f(x_t)$ in place of the true stochastic gradient $\widehat{\nabla} f(x_t)$ when computing the new iterate $x_{t+1}$.

Prior works such as [7] only consider the setting where the global minimizer lies in $\mathcal{X}$. Our algorithm and analysis does not require this restriction and instead only uses the following initial gradient estimate assumption from [22]:

**(5) Initial gradient estimate**: Let $x_1$ be the initial point. We assume that we have access to an upperbound $\nabla_1$ of $\|\nabla f(x_1)\|_*$ i.e. $\|\nabla f(x_1)\|_* \leq \nabla_1$. This assumption is justified as follows. If the noise parameter $\sigma$ defined in assumption (3) is known, we can use the procedure of [21] to estimate $\|\nabla f(x_1)\|_*$: we take $O(\ln(1/\delta))$ stochastic gradient samples at $x_1$, and let $g_1$ be the geometric median of these samples; we then set $\nabla_1 := \|g_1\|_* + 10\sigma$. It follows from [21] that $\|\nabla f(x_1)\|_* \leq \nabla_1$ holds with probability at least $1 - \delta$. If the domain contains the global optimum $x^*$ ($\nabla f(x^*) = 0$) and the initial distance $\|x_1 - x^*\|$ is known, we have the following alternative upper bound that follows from $\nabla f(x^*) = 0$ and smoothness: $\|\nabla f(x_1)\|_* = \|\nabla f(x_1) - \nabla f(x^*)\|_* \leq L\|x_1 - x^*\|$.

**Convergence guarantees**. We first state the convergence guarantee for this algorithm in Theorem 4.1 which works for an arbitrary norm and a general domain which may not include the global optimum. In this theorem, we assume that we know several problem parameters to show the main idea of our analysis. In Theorem 4.4, we remove the knowledge of the problem parameters.

**Theorem 4.1.** *Assume that convex $f$ satisfies Assumptions (1), (2), (3), (4) and (5). Let $\gamma = \max\{\log\frac{1}{\delta}; 1\}$; $R_1 = \sqrt{2\mathbf{D}_\psi(x^*, x_1)}$, and assume that $\nabla_1$ is an upper bound of $\|\nabla f(x_1)\|_*$. For known $T$, we choose $\lambda_t$ and $\eta_t$ such that*

$$\lambda_t = \lambda = \max\left\{\left(\frac{26T}{\gamma}\right)^{1/p} \sigma; 2(3LR_1 + \nabla_1)\right\}, \text{ and}$$

$$\eta_t = \eta = \frac{R_1}{24\lambda_t\gamma} = \frac{R_1}{24\gamma} \min\left\{\left(\frac{26T}{\gamma}\right)^{-1/p} \sigma^{-1}; \frac{1}{2}(3LR_1 + \nabla_1)^{-1}\right\}.$$

*Then with probability at least $1 - \delta$*

$$\frac{1}{T}\sum_{t=2}^{T+1} \Delta_t \leq 48R_1 \max\left\{26^{\frac{1}{p}} T^{\frac{1-p}{p}} \sigma\gamma^{\frac{p-1}{p}}; 2(3LR_1 + \nabla_1)T^{-1}\gamma\right\} = O\left(T^{\frac{1-p}{p}}\right).$$

*Remark* 4.2. This theorem shows that the convergence rate for the known time horizon case is $O(T^{\frac{1-p}{p}})$. This rate is known to be optimal, matching the lower bounds shown in [26, 31]. The above guarantee is also adaptive to $\sigma$, i.e., when $\sigma \to 0$, we obtain the standard $O(T^{-1})$ convergence rate of deterministic mirror descent.

*Remark* 4.3. The term $\nabla_1$ in the above theorem comes from the inexact estimation of $\|\nabla f(x_1)\|_*$. If we assume that the global optimum lies in the domain $\mathcal{X}$, we can simply select $\nabla_1 = LR_1$ without using the estimation procedure, as discussed in (5).

In Theorem 4.1, we use the initial distance $R_1$ to the optimal solution to set the step sizes and clipping parameters. This information is generally not available, but can be avoided. For example, for constrained problems where the domain radius is bounded by $R$, we can replace $R_1$ in Theorem 4.1 by $R$ without change in the dependency. However, for the general problem, we present Theorem 4.4, where we do not require knowledge of the constants $T, \sigma, \delta$ or $R_1$ to set the step sizes and clipping parameters. However, we still need the mild assumption of knowing an upper bound $\nabla_1$ on $\|\nabla f(x_1)\|_*$. As discussed in (5), $\nabla_1$ can be estimated with good accuracy when $\sigma$ is known.

**Theorem 4.4.** *Assume that convex $f$ satisfies Assumption (1), (2), (3), (4) and (5). Let $\gamma = \max\left\{\log\frac{1}{\delta}; 1\right\}$; $R_1 = \sqrt{2\mathbf{D}_\psi(x^*, x_1)}$, and assume that $\nabla_1$ is an upper bound of $\|\nabla f(x_1)\|_*$. We choose $\lambda_t$ and $\eta_t$ such that*

$$\lambda_t = \max\left\{\left(52t(1+\log t)^2 c_2\right)^{1/p}; 2\left(L\max_{i \le t}\|x_i - x_1\| + \nabla_1\right); \frac{Lc_1}{6}\right\}, \text{ and}$$

$$\eta_t = \frac{c_1}{24\lambda_t} = \frac{c_1}{24}\min\left\{\left(52t(1+\log t)^2 c_2\right)^{-1/p}; \frac{1}{2\left(L\max_{i \le t}\|x_i - x_1\| + \nabla_1\right)}; \frac{6}{Lc_1}\right\},$$

*where the absolute constants $c_1$ and $c_2$ are to ensure the correctness of the dimensions. Then, with probability at least $1 - \delta$, we have*

$$\frac{1}{T}\sum_{t=2}^{T+1}\Delta_t \le \frac{8}{Tc_1}\left(R_1 + \frac{c_1}{3}\left(\gamma + \frac{2\sigma^p}{c_2}\right)\right)^2 \max\left\{\left(52T(1+\log T)^2 c_2\right)^{1/p};\right.$$

$$\left. 4R_1 L + \frac{2c_1}{3}L\left(\gamma + \frac{2\sigma^p}{c_2}\right) + 2\nabla_1; \frac{Lc_1}{6}\right\} = \widetilde{O}\left(T^{\frac{1-p}{p}}\right).$$

**Sketch of the analysis**. In the remainder of this section, we provide a sketch of the analysis for Theorem 4.1, which starts with the following lemma.

**Lemma 4.5.** *Assume that convex $f$ satisfies Assumption (1), (2), (3), (4) and $\eta_t \le \frac{1}{4L}$, the iterate sequence $(x_t)_{t \ge 1}$ output by Algorithm 2 satisfies the following:*

$$\eta_t\Delta_{t+1} \le \mathbf{D}_\psi(x^*, x_t) - \mathbf{D}_\psi(x^*, x_{t+1}) + \eta_t\langle x^* - x_t, \theta_t^u\rangle + \eta_t\langle x^* - x_t, \theta_t^b\rangle$$

$$+ 2\eta_t^2\left(\|\theta_t^u\|_*^2 - \mathbb{E}\left[\|\theta_t^u\|_*^2 \mid \mathcal{F}_{t-1}\right]\right) + 2\eta_t^2\mathbb{E}\left[\|\theta_t^u\|_*^2 \mid \mathcal{F}_{t-1}\right] + 2\eta_t^2\|\theta_t^b\|_*^2.$$

*Remark* 4.6. In contrast to Remark 3.4, there is a mismatch between the gain $\Delta_{t+1}$ and the loss $\langle x^* - x_t, \theta_t\rangle$. Since the distance $\|x^* - x_t\|$ and the function value gap $\Delta_t$ cannot be related in the general convex case, we do not obtain the same rate as in the nonconvex case.

We now define the following terms for $t \ge 1$:

$$Z_t := z_t\left(\eta_t\Delta_{t+1} + \mathbf{D}_\psi(x^*, x_{t+1}) - \mathbf{D}_\psi(x^*, x_t) - \eta_t\langle x^* - x_t, \theta_t^b\rangle - 2\eta_t^2\|\theta_t^b\|_*^2\right.$$

$$\left. - 2\eta_t^2\mathbb{E}\left[\|\theta_t^u\|_*^2 \mid \mathcal{F}_{t-1}\right]\right) - \left(\frac{3}{8\lambda_t^2} + 24z_t^2\eta_t^4\lambda_t^2\right)\mathbb{E}\left[\|\theta_t^u\|^2 \mid \mathcal{F}_{t-1}\right],$$

where $z_t := \dfrac{1}{2\eta_t\lambda_t\max_{i \le t}\sqrt{2\mathbf{D}_\psi(x^*, x_i)} + 16Q\eta_t^2\lambda_t^2}$

for a constant $Q \ge 1$. We also define $S_t := \sum_{i=1}^{t} Z_i$. We have the following lemma, which is analogous to Lemma 3.6 in the nonconvex case.

**Lemma 4.7.** *For any $\delta > 0$, let $E(\delta)$ be the event that for all $1 \le k \le T$*

$$\sum_{t=1}^{k}z_t\eta_t\Delta_{t+1} + z_k\mathbf{D}_\psi(x^*, x_{k+1}) \le z_1\mathbf{D}_\psi(x^*, x_1) + \log\frac{1}{\delta} + \sum_{t=1}^{k}z_t\eta_t\langle x^* - x_t, \theta_t^b\rangle$$

$$+ 2\sum_{t=1}^{k}z_t\eta_t^2\|\theta_t^b\|_*^2 + \sum_{t=1}^{k}\left(\left(2z_t\eta_t^2 + \frac{3}{8\lambda_t^2} + 24z_t^2\eta_t^4\lambda_t^2\right)\mathbb{E}\left[\|\theta_t^u\|_*^2 \mid \mathcal{F}_{t-1}\right]\right).$$

$$(6)$$

*Then $\Pr[E(\delta)] \ge 1 - \delta$.*

---
**Algorithm 3** Clipped-ASMD
---
Parameters: initial point $y_1 = z_1$, step sizes $\{\eta_t\}$, clipping parameters $\{\lambda_t\}$, and mirror map $\psi$, where $\psi$ is 1-strongly convex wrt $\|\cdot\|$.

For $t = 1$ to $T$ do:

    Set $\alpha_t = \frac{2}{t+1}$.

    $x_t = (1 - \alpha_t)\, y_t + \alpha_t z_t$.

    $\widetilde{\nabla} f(x_t) = \min \left\{ 1, \frac{\lambda_t}{\|\widehat{\nabla} f(x_t)\|_*} \right\} \widehat{\nabla} f(x_t)$.

    $z_{t+1} = \arg\min_{x \in \mathcal{X}} \left\{ \eta_t \left\langle \widetilde{\nabla} f(x_t), x \right\rangle + \mathbf{D}_\psi (x, z_t) \right\}$.

    $y_{t+1} = (1 - \alpha_t)\, y_t + \alpha_t z_{t+1}$.

---

We now specify the choice of $\eta_t$ and $\lambda_t$. The following proposition gives a general condition for the choice of $\eta_t$ and $\lambda_t$ that gives the right convergence rate in time $T$.

**Proposition 4.8.** *We assume that the event $E(\delta)$ from Lemma 4.7 happens. Suppose that for some $\ell \leq T$, there are constants $C_1, C_2, C_3$, and $A$ such that for all $t \leq \ell$*

*1. $\lambda_t \eta_t = C_1$;  2. $\sum_{t=1}^{\ell} \left( \frac{1}{\lambda_t} \right)^p \leq C_2$;  3. $\left( \frac{1}{\lambda_t} \right)^{2p} \leq C_3 \left( \frac{1}{\lambda_t} \right)^p$;  4. $\|\nabla f(x_t)\|_* \leq \frac{\lambda_t}{2}$.*

*Then for all $t \leq \ell + 1$*

$$\sum_{i=1}^{t} \eta_i \Delta_{i+1} + \mathbf{D}_\psi (x^*, x_{t+1}) \leq \frac{1}{2} (R_1 + 8AC_1)^2$$

*for $A \geq \max \left\{ \log \frac{1}{\delta} + 26\sigma^p C_2 + \frac{2\sigma^{2p} C_2 C_3}{A}; 1 \right\}$.*

Theorem 4.1 follows from Proposition 4.8. Both proofs can be found in the Appendix.

## 5   Accelerated Stochastic Mirror Descent and Extensions

In Section D in the Appendix, we also show the convergence and its analysis for Clipped Accelerated Stochastic Mirror Descent (Algorithm 3). We require the following additional assumption:

**(5') Global minimizer**: We assume that $\nabla f(x^*) = 0$.

In other words, we assume that the global minimizer lies in the domain of the problem. This assumption is consistent with the works of [7, 28]. Our analysis readily extends to non-smooth settings, and more generally to functions that satisfy $f(y) - f(x) \leq \langle \nabla f(x), y - x \rangle + G \|y - x\| + \frac{L}{2} \|y - x\|^2$, $\quad \forall y, x \in \mathcal{X}$. This condition is satisfied by both Lipschitz functions (when $L = 0$) and smooth functions (when $G = 0$). The key step is to extend Lemma 4.5. The proof follows from [15] and can be found in the Appendix.

## 6   Conclusion

In this work, we propose a new approach to design and analyze various clipped gradient algorithms in the presence of heavy-tailed noise. Our analysis applies to various standard settings, including Clipped-SMD and accelerated Clipped-SMD for convex objectives with general domains and Clipped-SGD for nonconvex objectives, and gives optimal high probability rates in all settings. Our algorithms allow for setting step-sizes and clipping parameters when the time horizon and problem parameters such as the initial distance are unknown. For future work, since our algorithms have the limitation of still requiring the knowledge of parameters like $L$ and $p$, it is of great interest to investigate the existence of a *fully-adaptive* method, like Adagrad, that converges under heavy-tailed noise without requiring the knowledge of *any* problem parameter. Finally, it would be interesting to extend our techniques to the setting of variational inequalities under heavy-tailed noise [6].

## Acknowledgement

TDN and AE were supported in part by NSF CAREER grant CCF-1750333, NSF grant III-1908510, and an Alfred P. Sloan Research Fellowship. THN and HN were supported by NSF CAREER grant CCF-1750716 and NSF grant 2311649.

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

# A  Freedman's inequality

**Lemma A.1** (Freedman's inequality). *Let $(X_t)_{t\geq 1}$ be a martingale difference sequence. Assume that there exists a constant $c > 0$ such that $|X_t| \leq c$ almost surely for all $t \geq 1$ and define $\sigma_t^2 = \mathbb{E}\left[X_t^2 \mid X_{t-1}, \ldots, X_1\right]$. Then for all $b > 0$, $F > 0$ and $T \geq 1$*

$$\Pr\left[\left|\sum_{t=1}^{T} X_t\right| > b \text{ and } \sum_{t=1}^{T} \sigma_t^2 \leq F\right] \leq 2\exp\left(-\frac{b^2}{2F + 2cb/3}\right).$$

# B  Missing Proofs from Section 3

*Proof of Lemma 3.3.* By the smoothness of $f$ and the update $x_{t+1} = x_t - \frac{1}{\eta_t}\widetilde{\nabla}f(x_t)$ we have

$$
\begin{aligned}
&f(x_{t+1}) - f(x_t) \\
&\leq \langle \nabla f(x_t), x_{t+1} - x_t\rangle + \frac{L}{2}\|x_{t+1} - x_t\|^2 \\
&= -\eta_t\left\langle \nabla f(x_t), \widetilde{\nabla}f(x_t)\right\rangle + \frac{L\eta_t^2}{2}\left\|\widetilde{\nabla}f(x_t)\right\|^2 \\
&= -\eta_t\langle \nabla f(x_t), \theta_t + \nabla f(x_t)\rangle + \frac{L\eta_t^2}{2}\|\theta_t + \nabla f(x_t)\|^2 \\
&= -\eta_t\|\nabla f(x_t)\|^2 - \eta_t\langle \nabla f(x_t), \theta_t\rangle + \frac{L\eta_t^2}{2}\|\theta_t\|^2 + \frac{L\eta_t^2}{2}\|\nabla f(x_t)\|^2 + L\eta_t^2\langle \nabla f(x_t), \theta_t\rangle \\
&= -\left(\eta_t - \frac{L\eta_t^2}{2}\right)\|\nabla f(x_t)\|^2 + \frac{L\eta_t^2}{2}\|\theta_t\|^2 + \left(L\eta_t^2 - \eta_t\right)\langle \nabla f(x_t), \theta_t\rangle \\
&= -\left(\eta_t - \frac{L\eta_t^2}{2}\right)\|\nabla f(x_t)\|^2 + \frac{L\eta_t^2}{2}\|\theta_t\|^2 + \underbrace{\left(L\eta_t^2 - \eta_t\right)}_{\leq 0}\langle \nabla f(x_t), \theta_t^u + \theta_t^b\rangle.
\end{aligned}
$$

Using Cauchy-Schwarz, we have $\langle \nabla f(x_t), \theta_t^b\rangle \leq \frac{1}{2}\|\nabla f(x_t)\|^2 + \frac{1}{2}\|\theta_t^b\|^2$. Thus, we derive

$$
\begin{aligned}
\Delta_{t+1} - \Delta_t &\leq -\left(\frac{2\eta_t - L\eta_t^2}{2}\right)\|\nabla f(x_t)\|^2 + \frac{L\eta_t^2}{2}\|\theta_t\|^2 + \left(L\eta_t^2 - \eta_t\right)\langle \nabla f(x_t), \theta_t^u\rangle \\
&\quad + \frac{\eta_t - L\eta_t^2}{2}\|\nabla f(x_t)\|^2 + \frac{\eta_t - L\eta_t^2}{2}\|\theta_t^b\|^2 \\
&\leq -\frac{\eta_t}{2}\|\nabla f(x_t)\|^2 + \frac{L\eta_t^2}{2}\|\theta_t\|^2 + \left(L\eta_t^2 - \eta_t\right)\langle \nabla f(x_t), \theta_t^u\rangle + \frac{\eta_t}{2}\|\theta_t^b\|^2 \\
&\leq -\frac{\eta_t}{2}\|\nabla f(x_t)\|^2 + L\eta_t^2\|\theta_t^u\|^2 + \left(L\eta_t^2 - \eta_t\right)\langle \nabla f(x_t), \theta_t^u\rangle + \left(L\eta_t^2 + \frac{\eta_t}{2}\right)\|\theta_t^b\|^2 \\
&\leq -\frac{\eta_t}{2}\|\nabla f(x_t)\|^2 + L\eta_t^2\|\theta_t^u\|^2 + \left(L\eta_t^2 - \eta_t\right)\langle \nabla f(x_t), \theta_t^u\rangle + \frac{3\eta_t}{2}\|\theta_t^b\|^2,
\end{aligned}
$$

where the third inequality is due to $\|\theta_t\|^2 \leq 2\|\theta_t^u\|^2 + 2\|\theta_t^b\|^2$, and the last inequality is due to $\eta_t \leq \frac{1}{L}$. Rearranging, adding, and subtracting $\mathbb{E}\left[\|\theta_t^u\|^2 \mid \mathcal{F}_{t-1}\right]$, we obtain the lemma. $\qquad\square$

*Detailed proof of Lemma 3.5.* We state the following simple properties of the choice of $\eta$ and $\lambda$ in Theorem 3.1. We have

$$\frac{1}{L}\left(\frac{\sigma}{\lambda}\right)^p \leq \eta \tag{7}$$

$$\eta \leq \frac{1}{L} \tag{8}$$

$$\left(\frac{\sigma}{\lambda}\right)^p T^{\frac{p}{3p-2}} \leq \frac{1}{32} \tag{9}$$

$$TL\left(\frac{\sigma}{\lambda}\right)^p \lambda^2\eta^2 \leq \frac{\Delta_1}{2048}. \tag{10}$$

We will now prove by induction on $N$ that $E_N$ happens with probability at least $1 - \frac{(N-1)\delta}{T}$. For $N = 1$, the event happens with probability 1. Suppose that for some $N \leq T$, $\Pr[E_N] \geq 1 - \frac{(N-1)\delta}{T}$. We will prove that $\Pr[E_{N+1}] \geq 1 - \frac{N\delta}{T}$.

Since the LHS of (5) is non-negative, for $k \leq N$, we have, under the event $E_N$,

$$\Delta_k \leq \Delta_1 + \left(L\eta^2 - \eta\right)\sum_{t=1}^{k-1}\langle\nabla f(x_t), \theta_t^u\rangle + L\eta^2\sum_{t=1}^{k-1}\left(\|\theta_t^u\|^2 - \mathbb{E}_t\left[\|\theta_t^u\|^2\right]\right)$$

$$+ \frac{3\eta}{2}\sum_{t=1}^{k-1}\|\theta_t^b\|^2 + L\eta^2\sum_{t=1}^{k-1}\mathbb{E}_t\left[\|\theta_t^u\|^2\right] \leq 2\Delta_1.$$

From the induction hypothesis and Lemma 3.3, we have that for all $k \leq N$, $\Delta_k \leq 2\Delta_1$. Since the LHS of (5) is non-negative, by summing over $t$ from 1 to $N$ we have,

$$\Delta_{N+1} \leq \underbrace{\left(\eta - L\eta^2\right)\sum_{t=1}^{N}\langle-\nabla f(x_t), \theta_t^u\rangle}_{A} + \underbrace{\frac{3\eta}{2}\sum_{t=1}^{N}\|\theta_t^b\|^2}_{B}$$

$$+ \underbrace{L\eta^2\sum_{t=1}^{N}\left(\|\theta_t^u\|^2 - \mathbb{E}_t\left[\|\theta_t^u\|^2\right]\right)}_{C} + \underbrace{L\eta^2\sum_{t=1}^{N}\mathbb{E}_t\left[\|\theta_t^u\|^2\right]}_{D}.$$

The bounds for $B$ and $D$ are straightforward from Lemma 2.1. First, with probability 1, we have $\|\theta_t^u\| \leq 2\lambda$. By the smoothness of $f$ and the fact that $f$ is bounded below, we have

$$\|\nabla f(x_t)\| \leq \sqrt{2L\Delta_t}.$$

Furthermore, when the event $E_N$ happens, we have

$$\|\nabla f(x_t)\| \leq \sqrt{2L\Delta_t} \leq \sqrt{4L\Delta_1} \leq \frac{\lambda}{2}.$$

Thus, we can apply Lemma 2.1 and obtain $\|\theta_t^b\| \leq 4\sigma^p\lambda^{1-p}$ and $\mathbb{E}_t\left[\|\theta_t^u\|^2\right] \leq 40\sigma^p\lambda^{2-p}$.

**Upperbound for $B$.** By (3), when the event $E_N$ happens,

$$B = \frac{3\eta}{2}\|\theta_t^b\|^2 \leq \frac{3\eta}{2}\sum_{t=1}^{N}16\sigma^{2p}\lambda^{2-2p} = 24\sigma^{2p}\lambda^{2-2p}\eta N$$

$$\leq 24T\left(\frac{\sigma}{\lambda}\right)^{2p}\lambda^2\eta \leq 24TL\left(\frac{\sigma}{\lambda}\right)^p\lambda^2\eta^2 \leq \frac{3\Delta_1}{256}.$$

**Upperbound for $D$.** By 4, when the event $E_N$ happens,

$$D = L\eta^2\sum_{t=1}^{N}\mathbb{E}_t\left[\|\theta_t^u\|^2\right] \leq L\eta^2\sum_{t=1}^{N}40\sigma^p\lambda^{2-p}$$

$$\leq 40\sigma^p\lambda^{2-p}L\eta^2 N \leq 40LT\left(\frac{\sigma}{\lambda}\right)^p(\lambda\eta)^2 \leq \frac{5\Delta_1}{256}.$$

To bound $A$ and $C$, we use Freedman's inequality (Theorem A.1). We define, for $t \geq 1$, the following random variables

$$Z_t = \begin{cases} -\nabla f(x_t) & \text{if } \Delta_t \leq 2\Delta_1 \\ 0 & \text{otherwise.} \end{cases}$$

Thus $\|Z_t\| \leq \|\nabla f(x_t)\| \leq 2\sqrt{L\Delta_1}$ for all $t$.

**Upperbound for $A$.** Instead of bounding $A = \left(\eta - L\eta^2\right) \sum_{t=1}^{N} \langle -\nabla f(x_t), \theta_t^u \rangle$, we will bound $A' = \left(\eta - L\eta^2\right) \sum_{t=1}^{N} \langle Z_t, \theta_t^u \rangle$. We check the conditions to apply Freedman's inequality. First $\mathbb{E}_t \left[ \left(\eta - L\eta^2\right) \langle Z_t, \theta_t^u \rangle \right] = 0$. Further, with probability 1, $\|\theta_t^u\|^2 \leq 2\lambda$, and $Z_t \leq 2\sqrt{L\Delta_1}$, thus $\left| \left(\eta - L\eta^2\right) \langle Z_t, \theta_t^u \rangle \right| \leq \left(\eta - L\eta^2\right) \|Z_t\| \|\theta_t^u\| \leq 4\sqrt{L\Delta_1} \left(\eta - L\eta^2\right) \lambda \leq 4\sqrt{L\Delta_1}\eta\lambda$. Hence, $\left\{ \left(\eta - L\eta^2\right) \langle Z_t, \theta_t^u \rangle \right\}$ is a bounded martingale difference sequence. Therefore, for constant $a$ and $F$ to be chosen we have

$$\Pr\left[ \left| \sum_{t=1}^{N} \left(\eta - L\eta^2\right) \langle Z_t, \theta_t^u \rangle \right| > a \text{ and } \sum_{t=1}^{N} \mathbb{E}_t \left[ \left( \left(\eta - L\eta^2\right) \langle Z_t, \theta_t^u \rangle \right)^2 \right] \leq F \ln \frac{4T}{\delta} \right]$$

$$\leq 2 \exp\left( -\frac{a^2}{2F \ln \frac{4T}{\delta} + \frac{8}{3}\sqrt{L\Delta_1}\eta\lambda a} \right)$$

We choose $a$ such that

$$2 \exp\left( -\frac{a^2}{2F \ln \frac{4T}{\delta} + \frac{8}{3}\sqrt{L\Delta_1}\eta\lambda a} \right) = \frac{\delta}{2T}$$

which gives

$$a = \left( \frac{4}{3}\sqrt{L\Delta_1}\eta\lambda + \sqrt{\frac{16L\Delta_1\eta^2\lambda^2}{9} + 2F} \right) \ln \frac{4T}{\delta}.$$

If we choose $F = 64L\Delta_1\sigma^p\lambda^{2-p}\eta^2 T$, we can easily show that $a \leq \frac{7\Delta_1}{12}$. Therefore, with probability at least $1 - \frac{\delta}{2T}$ we have

$$E_A = \left\{ \text{either } A' \leq \left| \sum_{t=1}^{N} \left(\eta - L\eta^2\right) \langle Z_t, \theta_t^u \rangle \right| \leq \frac{7\Delta_1}{12} \right.$$

$$\left. \text{or } \sum_{t=1}^{N} \mathbb{E}_t \left[ \left( \left(\eta - L\eta^2\right) \langle Z_t, \theta_t^u \rangle \right)^2 \right] > F \ln \frac{4T}{\delta} \right\}.$$

Also notice that under the event $E_N$, we have

$$\sum_{t=1}^{N} \mathbb{E}_t \left[ \left( \left(\eta - L\eta^2\right) \langle Z_t, \theta_t^u \rangle \right)^2 \right]$$

$$\leq \eta^2 \sum_{t=1}^{N} \mathbb{E}_t \left[ \|Z_t\|^2 \|\theta_t^u\|^2 \right] \leq 4\eta^2 L\Delta_1 \sum_{t=1}^{N} \mathbb{E}_t \left[ \|\theta_t^u\|^2 \right]$$

$$\leq 64L\Delta_1\sigma^p\lambda^{2-p}\eta^2 N \leq 64\Delta_1 LT \left( \frac{\sigma}{\lambda} \right)^p \lambda^2\eta^2 \leq F \leq F \ln \frac{4T}{\delta}. \tag{11}$$

Under $E_N$, we have that $Z_t = -\nabla f(x_t)$ for all $t \leq N$. Therefore, when $E_N \cap E_A$ happens, we have $A = A' \leq a$.

**Upperbound for $C$.** We check the conditions to apply Freedman's inequality. First, $\mathbb{E}_t \left[ L\eta^2 \left( \|\theta_t^u\|^2 - \mathbb{E}_t \left[ \|\theta_t^u\|^2 \right] \right) \right] = 0$. Further, with probability 1, $\|\theta_t^u\|^2 \leq 2\lambda$, thus $\left| L\eta^2 \left( \|\theta_t^u\|^2 - \mathbb{E}_t \left[ \|\theta_t^u\|^2 \right] \right) \right| \leq L\eta^2 \left( 4\lambda^2 + 4\lambda^2 \right) = 8L\lambda^2\eta^2$. Hence $\left\{ L\eta^2 \left( \|\theta_t^u\|^2 - \mathbb{E}_t \left[ \|\theta_t^u\|^2 \right] \right) \right\}$ is a bounded martingale difference sequence. Applying Freedman's inequality for constants $c$ and $G$ to be chosen, we have

$$\Pr\left[ \left| L\eta^2 \sum_{t=1}^{N} \left( \|\theta_t^u\|^2 - \mathbb{E}_t \left[ \|\theta_t^u\|^2 \right] \right) \right| > c \text{ and } \sum_{t=1}^{N} \mathbb{E}_t \left[ \left( L\eta^2 \left( \|\theta_t^u\|^2 - \mathbb{E}_t \left[ \|\theta_t^u\|^2 \right] \right) \right)^2 \right] \leq G \ln \frac{4T}{\delta} \right]$$

$$\leq 2 \exp\left( -\frac{c^2}{2G \ln \frac{4T}{\delta} + \frac{16}{3}L\lambda^2\eta^2 c} \right)$$

We choose $c$ such that

$$2\exp\left(-\frac{c^2}{2G\ln\frac{4T}{\delta}+\frac{16}{3}L\lambda^2\eta^2 c}\right)=\frac{\delta}{2T}$$

which gives

$$c=\left(\frac{8}{3}L\lambda^2\eta^2+\sqrt{\frac{64L^2\lambda^4\eta^4}{9}+2G}\right)\ln\frac{4T}{\delta}$$

If we choose $G=256L^2\sigma^p\lambda^{4-p}\eta^4 T$, a simple calculation shows that $c\leq\frac{7\Delta_1}{48}$. we can show that with probability at least $1-\frac{\delta}{2T}$, the following event happens

$$E_C=\left\{\text{either }C\leq\left|L\eta^2\sum_{t=1}^N\left(\|\theta_t^u\|^2-\mathbb{E}_t\left[\|\theta_t^u\|^2\right]\right)\right|\leq\frac{7\Delta_1}{48}\right.$$

$$\left.\text{or }\sum_{t=1}^N\mathbb{E}_t\left[\left(L\eta^2\left(\|\theta_t^u\|^2-\mathbb{E}_t\left[\|\theta_t^u\|^2\right]\right)\right)^2\right]\geq G\ln\frac{4T}{\delta}\right\}.$$

Notice that when $G=256L^2\sigma^p\lambda^{4-p}\eta^4 T$, under $E_N$ we have

$$\sum_{t=1}^N\mathbb{E}_t\left[\left(L\eta^2\left(\|\theta_t^u\|^2-\mathbb{E}_t\left[\|\theta_t^u\|^2\right]\right)\right)^2\right]$$

$$\leq 8L\lambda^2\eta^2\sum_{t=1}^N\mathbb{E}_t\left[\left|L\eta^2\left(\|\theta_t^u\|^2-\mathbb{E}_t\left[\|\theta_t^u\|^2\right]\right)\right|\right]\leq 16L^2\lambda^2\eta^4\sum_{t=1}^N\mathbb{E}\left[\|\theta_t^u\|^2\right]$$

$$\leq 256L^2\sigma^p\lambda^{4-p}\eta^4 N\leq G<G\ln\frac{4T}{\delta}. \tag{12}$$

Therefore, when $E_N\cap E_C$ happens, we have $C\leq c$.

Finally, combining all the bounds for $A,B,C,D$ using union bound and selecting $\lambda$ and $\eta$ appropriately to simplify the constants, we obtain the lemma. $\qquad\square$

*Proof of Lemma 3.6.* We have

$$\mathbb{E}\left[\exp\left(Z_t\right)\mid\mathcal{F}_{t-1}\right]\exp\left(\left(3z_t^2 L\eta_t^2\Delta_t+6L^2 z_t^2\eta_t^4\lambda_t^2\right)\mathbb{E}\left[\|\theta_t^u\|^2\mid\mathcal{F}_{t-1}\right]\right)$$

$$\overset{(a)}{\leq}\mathbb{E}\left[\exp\left(z_t\left((L\eta_t^2-\eta_t)\langle\nabla f(x_t),\theta_t^u\rangle+L\eta_t^2\left(\|\theta_t^u\|^2-\mathbb{E}\left[\|\theta_t^u\|^2\mid\mathcal{F}_{t-1}\right]\right)\right)\right)\mid\mathcal{F}_{t-1}\right]$$

$$\overset{(b)}{\leq}\exp\left(\mathbb{E}\left[\frac{3}{4}\left(z_t\left((L\eta_t^2-\eta_t)\langle\nabla f(x_t),\theta_t^u\rangle+L\eta_t^2\left(\|\theta_t^u\|^2-\mathbb{E}\left[\|\theta_t^u\|^2\mid\mathcal{F}_{t-1}\right]\right)\right)\right)^2\mid\mathcal{F}_{t-1}\right]\right)$$

$$\overset{(c)}{\leq}\exp\left(\mathbb{E}\left[\frac{3}{2}z_t^2\eta_t^2\|\nabla f(x_t)\|^2\|\theta_t^u\|^2\mid\mathcal{F}_{t-1}\right]+\mathbb{E}\left[\frac{3}{2}L^2 z_t^2\eta_t^4\|\theta_t^u\|^4\mid\mathcal{F}_{t-1}\right]\right)$$

$$\overset{(d)}{\leq}\exp\left(3z_t^2 L\eta_t^2\Delta_t\mathbb{E}\left[\|\theta_t^u\|^2\mid\mathcal{F}_{t-1}\right]+6L^2 z_t^2\eta_t^4\lambda_t^2\mathbb{E}\left[\|\theta_t^u\|^2\mid\mathcal{F}_{t-1}\right]\right)$$

$$=\exp\left(\left(3z_t^2 L\eta_t^2\Delta_t+6L^2 z_t^2\eta_t^4\lambda_t^2\right)\mathbb{E}\left[\|\theta_t^u\|^2\mid\mathcal{F}_{t-1}\right]\right)$$

For $(a)$ we use Lemma 3.3. For $(b)$ we use Lemma 2.2. Notice that

$$\mathbb{E}\left[\langle\nabla f(x_t),\theta_t^u\rangle\right]=\mathbb{E}\left[\|\theta_t^u\|_*^2-\mathbb{E}\left[\|\theta_t^u\|_*^2\mid\mathcal{F}_{t-1}\right]\right]=0,$$

and since $\|\theta_t^u\|\leq 2\lambda_t$ and $\|\nabla f(x_t)\|\leq\sqrt{2L\Delta_t}$ for an $L$-smooth function, we have

$$\left|(L\eta_t^2-\eta_t)\langle\nabla f(x_t),\theta_t^u\rangle+L\eta_t^2\left(\|\theta_t^u\|^2-\mathbb{E}\left[\|\theta^u\|^2\mid\mathcal{F}_{t-1}\right]\right)\right|$$

$$\leq 2\eta_t \lambda_t \|\nabla f(x_t)\| + L\eta_t^2 \left( \|\theta_t^u\|^2 + \mathbb{E}\left[ \|\theta^u\|^2 \mid \mathcal{F}_{t-1} \right] \right)$$
$$\leq 2\eta_t \lambda_t \|\nabla f(x_t)\| + 8L\eta_t^2 \lambda_t^2$$
$$\leq 2\eta_t \lambda_t \sqrt{2L\Delta_t} + 8L\eta_t^2 \lambda_t^2.$$

Thus $z_t \leq \frac{1}{2\eta_t \lambda_t \sqrt{2L\Delta_t} + 8L\eta_t^2 \lambda_t^2}$. For $(c)$ we use $(a+b)^2 \leq 2a^2 + 2b^2$ and $\mathbb{E}\left[ (X - \mathbb{E}\left[X\right])^2 \right] \leq \mathbb{E}\left[X^2\right]$. For $(d)$, we use $\|\nabla f(x_t)\|^2 \leq 2L\Delta_t$ and $\|\theta_t^u\| \leq 2\lambda_t$. We obtain

$$\mathbb{E}\left[ \exp\left(Z_t\right) \mid \mathcal{F}_{t-1} \right] \leq 1.$$

Therefore

$$\mathbb{E}\left[ \exp\left(S_t\right) \mid \mathcal{F}_{t-1} \right] = \exp\left(S_{t-1}\right) \mathbb{E}\left[ \exp\left(Z_t\right) \mid \mathcal{F}_{t-1} \right]$$
$$\leq \exp\left(S_{t-1}\right)$$

which means $(\exp\left(S_t\right))_{t \geq 1}$ is a supermartingale. By Ville's inequality, we have, for all $k \geq 1$

$$\Pr\left[ S_k \geq \log \frac{1}{\delta} \right] \leq \delta \mathbb{E}\left[ \exp\left(S_1\right) \right] \leq \delta.$$

In other words, with probability at least $1 - \delta$, for all $k \geq 1$

$$\sum_{t=1}^{k} Z_t \leq \log \frac{1}{\delta}.$$

Plugging in the definition of $Z_t$ we have

$$\frac{1}{2} \sum_{t=1}^{k} z_t \eta_t \|\nabla f(x_t)\|^2 + \sum_{t=1}^{k} \left( z_t \Delta_{t+1} - z_t \Delta_t \right)$$

$$\leq \log \frac{1}{\delta} + \sum_{t=1}^{k} \frac{3z_t \eta_t}{2} \|\theta_t^b\|^2$$

$$+ \sum_{t=1}^{k} \left( \left( 3z_t^2 L\eta_t^2 \Delta_t + 6L^2 z_t^2 \eta_t^4 \lambda_t^2 + z_t L\eta_t^2 \right) \mathbb{E}\left[ \|\theta_t^u\|^2 \mid \mathcal{F}_{t-1} \right] \right).$$

Note that we have $z_t$ is a decreasing sequence by construction (see the proof of Proposition 3.7 below). Hence, the LHS of the above inequality can be bounded by

$$\text{LHS} = \frac{1}{2} \sum_{t=1}^{k} z_t \eta_t \|\nabla f(x_t)\|^2 + z_k \Delta_{k+1} - z_1 \Delta_1 + \sum_{t=2}^{k} \left( z_{k-1} - z_k \right) \Delta_k$$

$$\geq \frac{1}{2} \sum_{t=1}^{k} z_t \eta_t \|\nabla f(x_t)\|^2 + z_k \Delta_{k+1} - z_1 \Delta_1.$$

We obtain the desired inequality. $\qquad \square$

*Proof of Proposition 3.7.* We will prove by induction on $k$ that

$$\frac{1}{2} \sum_{i=1}^{k} \eta_i \|\nabla f(x_i)\|^2 + \Delta_{k+1} \leq \left( \sqrt{\Delta_1} + 2\sqrt{A}C_1 \right)^2.$$

The base case $k = 0$ is trivial. Suppose the statement is true for all $t \leq k \leq \ell$. Now we show for $k + 1$. Recall that

$$z_t = \frac{1}{2P_t \eta_t \lambda_t \max_{i \leq t} \sqrt{2L\Delta_i} + 8Q_t L\eta_t^2 \lambda_t^2}.$$

Let us choose

$$P_t = \frac{C_1}{\lambda_t \eta_t \sqrt{2L}} \geq 1$$

$$Q_t = \frac{C_1^2 \sqrt{A}}{2L\eta_t^2 \lambda_t^2} \geq 1.$$

We have

$$z_t = \frac{1}{2C_1 \max_{i \leq t} \sqrt{\Delta_i} + 4C_1^2 \sqrt{A}}.$$

Now, note that $(z_t)_{t \geq 1}$ is a decreasing sequence. By the induction hypothesis $\max_{i \leq k} \sqrt{\Delta_i} \leq \sqrt{\Delta_1} + 2\sqrt{A}C_1$. Hence:

$$\frac{z_t}{z_k} = \frac{2C_1 \max_{i \leq k} \sqrt{\Delta_i} + 4C_1^2 \sqrt{A}}{2C_1 \max_{i \leq t} \sqrt{\Delta_i} + 4C_1^2 \sqrt{A}}$$

$$\leq \frac{2C_1 \left(\sqrt{\Delta_1} + 2\sqrt{A}C_1\right) + 4C_1^2 \sqrt{A}}{2C_1 \sqrt{\Delta_1} + 4C_1^2 \sqrt{A}}$$

$$= \frac{\sqrt{\Delta_1} + 4\sqrt{A}C_1}{\sqrt{\Delta_1} + 2\sqrt{A}C_1} \leq 2.$$

By the choice of $\lambda_t$, for all $t \leq k$, $\|\nabla f(x_t)\| \leq \frac{\lambda_t}{2}$, we can apply the second part of Lemma 2.1 to obtain

$$\|\theta_t^b\| \leq 4\sigma^p \lambda_t^{1-p};$$

$$\mathbb{E}\left[\|\theta_t^u\|^2 \mid \mathcal{F}_{t-1}\right] \leq 40\sigma^p \lambda_t^{2-p}.$$

Thus,

$$\frac{1}{2} z_k \sum_{t=1}^{k} \eta_t \|\nabla f(x_t)\|^2 + z_k \Delta_{k+1}$$

$$\leq z_1 \Delta_1 + \log \frac{1}{\delta} + \sum_{t=1}^{k} \frac{3z_t \eta_t}{2} \|\theta_t^b\|^2$$

$$+ \sum_{t=1}^{k} \left(\left(3z_t^2 L\eta_t^2 \Delta_t + 6L^2 z_t^2 \eta_t^4 \lambda_t^2 + z_t L\eta_t^2\right) \mathbb{E}\left[\|\theta_t^u\|^2 \mid \mathcal{F}_{t-1}\right]\right)$$

$$\leq z_1 \Delta_1 + \log \frac{1}{\delta} + 24\sigma^{2p} \sum_{t=1}^{k} z_t \eta_t \lambda_t^2 \left(\frac{1}{\lambda_t}\right)^{2p}$$

$$+ 40\sigma^p \sum_{t=1}^{k} \left(\left(3z_t^2 \Delta_t + 6z_t^2 L\eta_t^2 \lambda_t^2 + z_t\right) L\eta_t^2 \lambda_t^2 \left(\frac{1}{\lambda_t}\right)^p\right).$$

Since $\frac{z_t}{z_k} \leq 2$, we have

$$\frac{1}{2} \sum_{t=1}^{k} \eta_t \|\nabla f(x_t)\|^2 + \Delta_{k+1}$$

$$\leq \frac{z_1 \Delta_1}{z_k} + \frac{1}{z_k} \log \frac{1}{\delta} + 48\sigma^{2p} \sum_{t=1}^{k} \eta_t \lambda_t^2 \left(\frac{1}{\lambda_t}\right)^{2p}$$

$$+ 80\sigma^p \sum_{t=1}^{k} \left(\left(3z_t \Delta_t + 6z_t L\eta_t^2 \lambda_t^2 + 1\right) L\eta_t^2 \lambda_t^2 \left(\frac{1}{\lambda_t}\right)^p\right)$$

$$\overset{(a)}{\leq} \frac{\sqrt{\Delta_1} + 4\sqrt{A}C_1}{\sqrt{\Delta_1} + 2\sqrt{A}C_1} \Delta_1 + 2C_1 \left(\sqrt{\Delta_1} + 4\sqrt{A}C_1\right) \log \frac{1}{\delta} + 48\sigma^{2p} C_2 \sum_{t=1}^{k} L\eta_t^2 \lambda_t^2 \left(\frac{1}{\lambda_t}\right)^p$$

$$+ 80\sigma^p \sum_{t=1}^{k} \left(\left(\frac{3\left(\sqrt{\Delta_1} + 2\sqrt{A}C_1\right)^2}{2C_1 \left(\sqrt{\Delta_1} + 2\sqrt{A}C_1\right)} + \frac{6}{8Q_t} + 1\right) L\eta_t^2 \lambda_t^2 \left(\frac{1}{\lambda_t}\right)^p\right)$$

$$\overset{(b)}{\leq} \Delta_1 + 2\sqrt{\Delta_1}\sqrt{A}C_1 + 2C_1\left(\sqrt{\Delta_1} + 4\sqrt{A}C_1\right)\log\frac{1}{\delta} + 48\sigma^{2p}C_2C_3$$

$$+ 80\sigma^p\left(\frac{3\left(\sqrt{\Delta_1} + 2\sqrt{A}C_1\right)}{2C_1} + \frac{7}{4}\right)C_3$$

$$\leq \Delta_1 + 2\sqrt{\Delta_1}\sqrt{A}C_1 + 2C_1\left(\sqrt{\Delta_1} + 4\sqrt{A}C_1\right)\left(\log\frac{1}{\delta} + \frac{60\sigma^p C_3}{C_1^2}\right)$$

$$+ 48\sigma^{2p}C_2C_3 + 140\sigma^p C_3$$

$$\overset{(c)}{\leq} \Delta_1 + 2\sqrt{\Delta_1}\sqrt{A}C_1 + 2C_1\left(\sqrt{\Delta_1} + 4\sqrt{A}C_1\right)\frac{\sqrt{A}}{8} + AC_1^2$$

$$\leq \left(\sqrt{\Delta_1} + 2\sqrt{A}C_1\right)^2.$$

For $(a)$, we use $\left(\frac{1}{\lambda_t}\right)^p \leq C_2 L\eta_t$ and the induction hypothesis. For $(b)$, we use $\sum_{t=1}^{T} L\left(\frac{1}{\lambda_t}\right)^p \lambda_t^2\eta_t^2 \leq C_3$ and $Q_t \geq 1$. For $(c)$, we have

$$\log\frac{1}{\delta} + \frac{60\sigma^p C_3}{C_1^2} \leq \frac{\sqrt{A}}{8}$$

$$48\sigma^{2p}C_2C_3 + 140\sigma^p C_3 \leq AC_1^2,$$

since

$$A \geq 64\left(\log\frac{1}{\delta} + \frac{60\sigma^p C_3}{C_1^2}\right)^2 + \frac{48\sigma^{2p}C_2C_3 + 140\sigma^p C_3}{C_1^2}.$$

This concludes the proof. □

**Lemma B.1.** *The choices of $\eta_t$ and $\lambda_t$ in Theorem 3.1 satisfy the condition (1)-(3) of Proposition 3.7 for*

$$C_1 = \frac{\sqrt{\Delta_1}}{4\sqrt{2}\gamma},$$

$$C_2 = \frac{1}{\sigma^p},$$

$$C_3 = \frac{\Delta_1}{2048\sigma^p\gamma}.$$

*Proof.* We verify for the first case. The second follows exactly the same. First, we have $p > 1$ hence

$$\eta_t\lambda_t\sqrt{2L} = \frac{\sqrt{\Delta_1}T^{\frac{1-p}{3p-2}}}{8\sqrt{L}\gamma}\sqrt{2L} \leq \frac{\sqrt{\Delta_1}}{4\sqrt{2}\gamma} = C_1.$$

Since $\eta_t = \frac{\sqrt{\Delta_1}T^{\frac{1-p}{3p-2}}}{8\lambda_t\sqrt{L}\gamma}$, $p > 1$ and $\lambda_t \geq \left(\frac{8\gamma}{\sqrt{L\Delta_1}}\right)^{\frac{1}{p-1}}T^{\frac{1}{3p-2}}\sigma^{\frac{p}{p-1}}$

$$\eta_t\lambda_t^p = \frac{\sqrt{\Delta_1}T^{\frac{1-p}{3p-2}}}{8\sqrt{L}\gamma}\lambda_t^{p-1}$$

$$\geq \frac{\sqrt{\Delta_1}T^{\frac{1-p}{3p-2}}}{8\sqrt{L}\gamma}\frac{8\gamma}{\sqrt{L\Delta_1}}T^{\frac{p-1}{3p-2}}\sigma^p$$

$$= \frac{\sigma^p}{L}$$

which gives

$$\frac{1}{L\eta_t}\left(\frac{1}{\lambda_t}\right)^p \leq \frac{1}{\sigma^p} = C_2.$$

Finally, we have $\lambda_t \geq 32^{1/p}\sigma T^{\frac{1}{3p-2}}$ hence

$$\left(\frac{1}{\lambda_t}\right)^p T^{\frac{p}{3p-2}} \leq \frac{1}{32\sigma^p}.$$

Therefore,

$$
\begin{aligned}
\sum_{t=1}^{T} L\left(\frac{1}{\lambda_t}\right)^p \lambda_t^2 \eta_t^2 &= \sum_{t=1}^{T} L\left(\frac{1}{\lambda_t}\right)^p \left(\frac{\sqrt{\Delta_1}T^{\frac{1-p}{3p-2}}}{8\sqrt{L}\gamma}\right)^2 \\
&= \frac{1}{T}\sum_{t=1}^{T} L\left(\frac{1}{\lambda_t}\right)^p T \cdot T^{\frac{2-2p}{3p-2}}\frac{\Delta_1}{64L\gamma} \\
&= \frac{1}{T}\sum_{t=1}^{T} \left(\frac{1}{\lambda_t}\right)^p T^{\frac{p}{3p-2}}\frac{\Delta_1}{64\gamma^2} \\
&\leq \frac{1}{T}\sum_{t=1}^{T} \frac{1}{32\sigma^p}\frac{\Delta_1}{64\gamma^2} \\
&= \frac{1}{32\sigma^p}\frac{\Delta_1}{64\gamma^2} \leq \frac{\Delta_1}{2048\sigma^p\gamma}.
\end{aligned}
$$

$\square$

*Proof of Theorem 3.1.* Note that $\eta \leq \frac{T^{\frac{1-p}{3p-2}}}{16\sqrt{90}L\gamma} \leq \frac{1}{L}$. We have that with probability at least $1 - \delta$, event $E(\delta)$ happens. Conditioning on this event, we verify the conditions of Proposition 3.7. We select the following constants

$$C_1 = \frac{\sqrt{\Delta_1}}{4\sqrt{2}\gamma}; \quad C_2 = \frac{1}{\sigma^p}; \quad C_3 = \frac{\Delta_1}{2048\sigma^p\gamma}; \quad A = 256\gamma^2.$$

We verify in Lemma B.1 that for these choice of constants, conditions (1)-(3) of Proposition 3.7 are satisfied. Furthermore, we have

$$
\begin{aligned}
64&\left(\log\frac{1}{\delta} + \frac{60\sigma^p C_3}{C_1^2}\right)^2 + \frac{48\sigma^{2p}C_2C_3 + 140\sigma^p C_3}{C_1^2} \\
&= 64\left(\log\frac{1}{\delta} + 60\log\frac{1}{\delta}\frac{32}{\Delta_1}\frac{\Delta_1}{2048}\right)^2 + \left(48\frac{\Delta_1}{2048} + 140\frac{\Delta_1}{2048}\right)\frac{32}{\Delta_1} \\
&\leq 256\gamma^2 = A.
\end{aligned}
$$

We only need to show that, for all $t$, $\|\nabla f(x_t)\| \leq \frac{\lambda_t}{2}$. We will show this by induction. Indeed, for the base case we have $\|\nabla f(x_1)\| \leq \sqrt{2L\Delta_1} \leq \frac{\lambda_1}{2}$. Suppose that it is true for all $t \leq k$. We will prove that $\|\nabla f(x_{k+1})\| \leq \frac{\lambda_{k+1}}{2}$. By Proposition 3.7 and the induction hypothesis

$$\Delta_{k+1} \leq \left(\sqrt{\Delta_1} + 2\sqrt{A}C_1\right)^2 \leq \left(\sqrt{\Delta_1} + \frac{\sqrt{\Delta_1}}{2\sqrt{2}\gamma}\times 16\gamma\right)^2 \leq 45\Delta_1.$$

Thus, we get

$$\|\nabla f(x_{k+1})\| \leq \sqrt{2L\Delta_{k+1}} \leq \sqrt{90L\Delta_1} \leq \frac{\lambda_{k+1}}{2}$$

as needed. From Proposition 3.7, we have

$$\frac{\eta}{2}\sum_{t=1}^{T}\|\nabla f(x_t)\|^2 + \Delta_{k+1} \leq 45\Delta_1.$$

Therefore

$$\frac{1}{T}\sum_{t=1}^{T}\|\nabla f(x_t)\|^2 \leq \frac{90\Delta_1}{\eta T} = 720\sqrt{\Delta_1 L}\gamma\max\left\{\left(\frac{8\gamma}{\sqrt{L\Delta_1}}\right)^{\frac{1}{p-1}}T^{\frac{2-2p}{3p-2}}\sigma^{\frac{p}{p-1}}; 2\sqrt{90L\Delta_1}T^{\frac{1-2p}{3p-2}}; 32^{\frac{1}{p}}\sigma T^{\frac{2-2p}{3p-2}}\right\}.$$

$\square$

**Theorem B.2.** *Assume that $f$ satisfies Assumption (1'), (2), (3), (4). Let $\gamma = \max\left\{\log\frac{1}{\delta}; 1\right\}$ and $\Delta_1 = f(x_1) - f^*$. For unknown $T$, we choose $\lambda_t$ and $\eta_t$ such that*

$$\lambda_t = \max\left\{ \left(\frac{8\gamma}{\sqrt{L\Delta_1}}\right)^{\frac{1}{p-1}} \left(2t\left(1+\log t\right)^2\right)^{\frac{1}{3p-2}} \sigma^{\frac{p}{p-1}}; 2\sqrt{90L\Delta_1}; 32^{\frac{1}{p}}\sigma\left(2t\left(1+\log t\right)^2\right)^{\frac{1}{3p-2}} \right\},$$

$$\eta_t = \frac{\sqrt{\Delta_1}\left(2t\left(1+\log t\right)^2\right)^{\frac{1-p}{3p-2}}}{8\lambda_t\sqrt{L}\gamma}.$$

*Then with probability at least $1 - \delta$*

$$\frac{1}{T}\sum_{t=1}^{T}\|\nabla f(x_t)\|^2 \leq 720\sqrt{\Delta_1 L}\gamma \max\left\{ \left(\frac{8\gamma}{\sqrt{L\Delta_1}}\right)^{\frac{1}{p-1}} \left(2\left(1+\log T\right)^2\right)^{\frac{p}{3p-2}} \sigma^{\frac{p}{p-1}} T^{\frac{2-2p}{3p-2}}; \right.$$

$$\left. 2\sqrt{90L\Delta_1}\left(2\left(1+\log T\right)^2\right)^{\frac{p-1}{3p-2}} T^{\frac{1-2p}{3p-2}}; 32^{\frac{1}{p}}\sigma\left(2\left(1+\log T\right)^2\right)^{\frac{p}{3p-2}} T^{\frac{2-2p}{3p-2}} \right\}.$$

We again verify the conditions of Proposition 3.7 for the choices of $\eta_t$ and $\lambda_t$ in Theorem B.2.

**Lemma B.3.** *The choices of $\eta_t$ and $\lambda_t$ in Theorem B.2 satisfy the condition (1)-(3) of Proposition 3.7 for*

$$C_1 = \frac{\sqrt{\Delta_1}}{4\sqrt{2}\gamma},$$

$$C_2 = \frac{1}{\sigma^p},$$

$$C_3 = \frac{\Delta_1}{2048\sigma^p\gamma}.$$

The proof utilizes the following fact:

**Fact B.4.** *We have $\sum_{t=1}^{\infty} \frac{1}{2t(1+\log t)^2} < 1$.*

*Proof.* First, we have $p > 1$ hence

$$\eta_t\lambda_t\sqrt{2L} = \frac{\sqrt{\Delta_1}\left(2t\left(1+\log t\right)^2\right)^{\frac{1-p}{3p-2}}}{8\sqrt{L}\gamma}\sqrt{2L}$$

$$\leq \frac{\sqrt{\Delta_1}}{4\sqrt{2}\gamma} = C_1.$$

Since $\eta_t = \frac{\sqrt{\Delta_1}T^{\frac{1-p}{3p-2}}}{8\lambda_t\sqrt{L}\gamma}$, $p > 1$ and $\lambda_t \geq \left(\frac{8\gamma}{\sqrt{L\Delta_1}}\right)^{\frac{1}{p-1}}\left(2t\left(1+\log t\right)^2\right)^{\frac{1-p}{3p-2}}\sigma^{\frac{p}{p-1}}$

$$\eta_t\lambda_t^p = \frac{\sqrt{\Delta_1}\left(2t\left(1+\log t\right)^2\right)^{\frac{1-p}{3p-2}}}{8\sqrt{L}\gamma}\lambda_t^{p-1}$$

$$\geq \frac{\sqrt{\Delta_1}\left(2t\left(1+\log t\right)^2\right)^{\frac{1-p}{3p-2}}}{8\sqrt{L}\gamma}\frac{8\gamma}{\sqrt{L\Delta_1}}\left(2t\left(1+\log t\right)^2\right)^{\frac{p-1}{3p-2}}\sigma^p$$

$$= \frac{\sigma^p}{L},$$

which gives

$$\frac{1}{L\eta_t}\left(\frac{1}{\lambda_t}\right)^p \leq \frac{1}{\sigma^p} = C_2.$$

Finally, we have $\lambda_t \geq 32^{\frac{1}{p}} \sigma \left( 2t \left( 1 + \log t \right)^2 \right)^{\frac{1}{3p-2}}$, hence

$$\left( \frac{1}{\lambda_t} \right)^p \left( 2t \left( 1 + \log t \right)^2 \right)^{\frac{p}{3p-2}} \leq \frac{1}{32\sigma^p}. \tag{13}$$

Therefore,

$$
\begin{aligned}
\sum_{t=1}^{T} L \left( \frac{1}{\lambda_t} \right)^p \lambda_t^2 \eta_t^2 &= \sum_{t=1}^{T} L \left( \frac{1}{\lambda_t} \right)^p \left( 2t \left( 1 + \log t \right)^2 \right)^{\frac{2-2p}{3p-2}} \left( \frac{\sqrt{\Delta_1}}{8\sqrt{L}\gamma} \right)^2 \\
&= \sum_{t=1}^{T} L \frac{1}{2t \left( 1 + \log t \right)^2} \left( \frac{1}{\lambda_t} \right)^p \left( 2t \left( 1 + \log t \right)^2 \right)^{\frac{p}{3p-2}} \frac{\Delta_1}{64\gamma^2} \\
&\leq \sum_{t=1}^{T} L \frac{1}{2t \left( 1 + \log t \right)^2} \frac{1}{32\sigma^p} \frac{\Delta_1}{64\gamma^2} \qquad\qquad \text{(by (13))} \\
&= \frac{1}{32\sigma^p} \frac{\Delta_1}{64\gamma^2} \sum_{t=1}^{T} \frac{1}{2t \left( 1 + \log t \right)^2} \\
&\leq \frac{1}{32\sigma^p} \frac{\Delta_1}{64\gamma^2} \leq \frac{\Delta_1}{2048\sigma^p\gamma}. \qquad\qquad \text{(by Fact B.4)}
\end{aligned}
$$

$\square$

*Proof of Theorem B.2.* Note that

$$
\begin{aligned}
\eta_t &= \frac{\sqrt{\Delta_1} \left( 2t \left( 1 + \log t \right)^2 \right)^{\frac{1-p}{3p-2}}}{8\lambda_t \sqrt{L}\gamma} \\
&\leq \frac{\left( 2t \left( 1 + \log t \right)^2 \right)^{\frac{1-p}{3p-2}}}{16L\gamma\sqrt{90}} \\
&\leq \frac{1}{L}.
\end{aligned}
$$

Note that with Lemma B.3, verifying the conditions of Proposition 3.7 is identical to the proof of theorem 3.1. We have that with probability at least $1 - \delta$, event $E(\delta)$ from 3.7 happens. We have with probability at least $1 - \delta$:

$$\frac{1}{2} \sum_{t=1}^{T} \eta_t \left\| \nabla f(x_t) \right\|^2 + \Delta_{k+1} \leq 45\Delta_1.$$

Since $\eta_t$ is decreasing, we have

$$\frac{1}{T} \sum_{t=1}^{T} \left\| \nabla f(x_t) \right\|^2 \leq \frac{90\Delta_1}{T\eta_T}.$$

This means that

$$
\begin{aligned}
\frac{1}{T} \sum_{t=1}^{T} \left\| \nabla f(x_t) \right\|^2 &\leq 720 \sqrt{\Delta_1 L} \gamma \max \Bigg\{ \left( \frac{8\gamma}{\sqrt{L\Delta_1}} \right)^{\frac{1}{p-1}} \left( 2 \left( 1 + \log T \right)^2 \right)^{\frac{p}{3p-2}} \sigma^{\frac{p}{p-1}} T^{\frac{2-2p}{3p-2}}; \\
&\qquad 2\sqrt{90L\Delta_1} \left( 2 \left( 1 + \log T \right)^2 \right)^{\frac{p-1}{3p-2}} T^{\frac{1-2p}{3p-2}}; 32^{\frac{1}{p}} \sigma \left( 2 \left( 1 + \log T \right)^2 \right)^{\frac{p}{3p-2}} T^{\frac{2-2p}{3p-2}} \Bigg\}.
\end{aligned}
$$

$\square$

## C Missing Proofs from Section 4

**Lemma C.1.** *Suppose that $\eta_t \leq \frac{1}{4L}$ and assume $f$ satisfies Assumption (1), (2), (3) as well as the following condition*

$$f(y) - f(x) \leq \langle \nabla f(x), y - x \rangle + G \|y - x\| + \frac{L}{2} \|y - x\|^2, \quad \forall y, x \in \mathcal{X}. \tag{14}$$

*Then the iterate sequence $(x_t)_{t \geq 1}$ output by Algorithm 2 satisfies the following:*

$$\eta_t \Delta_{t+1} \leq \mathbf{D}_\psi \left( x^*, x_t \right) - \mathbf{D}_\psi \left( x^*, x_{t+1} \right) + \eta_t \left\langle x^* - x_t, \theta_t^u \right\rangle + \eta_t \left\langle x^* - x_t, \theta_t^b \right\rangle$$
$$+ 2\eta_t^2 \left( \|\theta_t^u\|_*^2 - \mathbb{E} \left[ \|\theta_t^u\|_*^2 \mid \mathcal{F}_{t-1} \right] \right) + 2\eta_t^2 \mathbb{E} \left[ \|\theta_t^u\|_*^2 \mid \mathcal{F}_{t-1} \right] + 2\eta_t^2 \|\theta_t^b\|_*^2 + 2G^2 \eta_t^2.$$

*Proof.* By condition (14) and convexity,

$$f\left( x_{t+1} \right) - f\left( x^* \right) \leq \underbrace{f\left( x_{t+1} \right) - f\left( x_t \right)}_{\text{condition (14)}} + \underbrace{f\left( x_t \right) - f\left( x^* \right)}_{\text{convexity}}$$

$$\leq \langle \nabla f\left( x_t \right), x_{t+1} - x_t \rangle + \frac{L}{2} \|x_t - x_{t+1}\|^2 + G \|x_t - x_{t+1}\| + \langle \nabla f\left( x_t \right), x_t - x^* \rangle$$

$$= \langle \nabla f\left( x_t \right), x_{t+1} - x^* \rangle + \frac{L}{2} \|x_t - x_{t+1}\|^2 + G \|x_t - x_{t+1}\|$$

$$= \langle \theta_t, x^* - x_{t+1} \rangle + \left\langle \widetilde{\nabla} f(x_t), x_{t+1} - x^* \right\rangle + \frac{L}{2} \|x_t - x_{t+1}\|^2 + G \|x_t - x_{t+1}\|.$$

By the optimality condition, we have

$$\left\langle \eta_t \widetilde{\nabla} f(x_t) + \nabla_x \mathbf{D}_\psi \left( x_{t+1}, x_t \right), x^* - x_{t+1} \right\rangle \geq 0$$

and thus

$$\left\langle \eta_t \widetilde{\nabla} f(x_t), x_{t+1} - x^* \right\rangle \leq \langle \nabla_x \mathbf{D}_\psi \left( x_{t+1}, x_t \right), x^* - x_{t+1} \rangle.$$

Note that

$$\langle \nabla_x \mathbf{D}_\psi \left( x_{t+1}, x_t \right), x^* - x_{t+1} \rangle = \langle \nabla \psi \left( x_{t+1} \right) - \nabla \psi \left( x_t \right), x^* - x_{t+1} \rangle$$
$$= \mathbf{D}_\psi \left( x^*, x_t \right) - \mathbf{D}_\psi \left( x_{t+1}, x_t \right) - \mathbf{D}_\psi \left( x^*, x_{t+1} \right).$$

Thus

$$\eta_t \left\langle \widetilde{\nabla} f(x_t), x_{t+1} - x^* \right\rangle \leq \mathbf{D}_\psi \left( x^*, x_t \right) - \mathbf{D}_\psi \left( x^*, x_{t+1} \right) - \mathbf{D}_\psi \left( x_{t+1}, x_t \right)$$

$$\leq \mathbf{D}_\psi \left( x^*, x_t \right) - \mathbf{D}_\psi \left( x^*, x_{t+1} \right) - \frac{1}{2} \|x_{t+1} - x_t\|^2,$$

where we have used that $\mathbf{D}_\psi \left( x_{t+1}, x_t \right) \geq \frac{1}{2} \|x_{t+1} - x_t\|^2$ by the strong convexity of $\psi$.

Combining the two inequalities, and using the assumption that $L\eta_t \leq \frac{1}{4}$, we obtain

$$\eta_t \Delta_{t+1} + \mathbf{D}_\psi \left( x^*, x_{t+1} \right) - \mathbf{D}_\psi \left( x^*, x_t \right)$$

$$\leq \eta_t \langle \theta_t, x^* - x_{t+1} \rangle + \frac{L\eta_t}{2} \|x_t - x_{t+1}\|^2 + G\eta_t \|x_t - x_{t+1}\| - \frac{1}{2} \|x_{t+1} - x_t\|^2$$

$$\leq \eta_t \langle \theta_t, x^* - x_t \rangle + \eta_t \langle \theta_t, x_t - x_{t+1} \rangle - \frac{3}{8} \|x_{t+1} - x_t\|^2 + G\eta_t \|x_t - x_{t+1}\|$$

$$\leq \eta_t \langle \theta_t, x^* - x_t \rangle + \eta_t^2 \|\theta_t\|_*^2 + 2G^2 \eta_t^2$$

$$\leq \eta_t \left\langle \theta_t^u + \theta_t^b, x^* - x_t \right\rangle + 2\eta_t^2 \|\theta_t^u\|_*^2 + 2\eta_t^2 \|\theta_t^b\|_*^2 + 2G^2 \eta_t^2.$$

This is what we want to show. $\qquad\qquad\qquad\qquad\qquad\qquad\qquad\qquad\qquad\qquad\qquad\qquad\square$

*Proof of Lemma 4.7.* We have

$$\mathbb{E}\left[\exp\left(Z_t\right) \mid \mathcal{F}_{t-1}\right] \times \exp\left(\left(\frac{3}{8\lambda_t^2} + 24z_t^2\eta_t^4\lambda_t^2\right)\mathbb{E}\left[\|\theta_t^u\|_*^2 \mid \mathcal{F}_{t-1}\right]\right)$$

$$\overset{(a)}{\leq} \mathbb{E}\left[\exp\left(z_t\left(\eta_t\langle x^* - x_t, \theta_t^u\rangle + 2\eta_t^2\left(\|\theta_t^u\|_*^2 - \mathbb{E}\left[\|\theta_t^u\|_*^2 \mid \mathcal{F}_{t-1}\right]\right)\right)\right) \mid \mathcal{F}_{t-1}\right]$$

$$\overset{(b)}{\leq} \exp\left(\mathbb{E}\left[\frac{3}{4}\left(z_t\left(\eta_t\langle x^* - x_t, \theta_t^u\rangle + 2\eta_t^2\left(\|\theta_t^u\|_*^2 - \mathbb{E}\left[\|\theta_t^u\|_*^2 \mid \mathcal{F}_{t-1}\right]\right)\right)\right)^2 \mid \mathcal{F}_{t-1}\right]\right)$$

$$\overset{(c)}{\leq} \exp\left(\left(\frac{3}{2}z_t^2\eta_t^2\|x^* - x_t\|^2\,\mathbb{E}\left[\|\theta_t^u\|_*^2 \mid \mathcal{F}_{t-1}\right] + 6z_t^2\eta_t^4\mathbb{E}\left[\|\theta_t^u\|_*^4 \mid \mathcal{F}_{t-1}\right]\right)\right)$$

$$\overset{(d)}{\leq} \exp\left(\left(\frac{3}{2}z_t^2\eta_t^2\|x^* - x_t\|^2 + 24z_t^2\eta_t^4\lambda_t^2\right)\mathbb{E}\left[\|\theta_t^u\|_*^2 \mid \mathcal{F}_{t-1}\right]\right)$$

$$\overset{(e)}{\leq} \exp\left(\left(\frac{3}{8\lambda_t^2} + 24z_t^2\eta_t^4\lambda_t^2\right)\mathbb{E}\left[\|\theta_t^u\|_*^2 \mid \mathcal{F}_{t-1}\right]\right).$$

For $(a)$, we use Lemma 4.5. For $(b)$, we use Lemma 2.2. Notice that

$$\mathbb{E}\left[\langle x^* - x_t, \theta_t^u\rangle\right] = \mathbb{E}\left[\|\theta_t^u\|_*^2 - \mathbb{E}\left[\|\theta_t^u\|_*^2 \mid \mathcal{F}_{t-1}\right]\right] = 0,$$

and since $\|\theta_t^u\|_* \leq 2\lambda_t$, we have

$$\left|\eta_t\langle x^* - x_t, \theta_t^u\rangle + 2\eta_t^2\left(\|\theta_t^u\|_*^2 - \mathbb{E}\left[\|\theta_t^u\|_*^2 \mid \mathcal{F}_{t-1}\right]\right)\right|$$

$$\leq \eta_t\|x^* - x_t\|\|\theta_t^u\|_* + 2\eta_t^2\left(\|\theta_t^u\|_*^2 + \mathbb{E}\left[\|\theta_t^u\|_*^2 \mid \mathcal{F}_{t-1}\right]\right)$$

$$\leq 2\eta_t\lambda_t\|x^* - x_t\| + 16\eta_t^2\lambda_t^2$$

$$\leq 2\eta_t\lambda_t\sqrt{2\mathbf{D}_\psi(x^*, x_t)} + 16\eta_t^2\lambda_t^2.$$

Thus, $z_t \leq \frac{1}{2\eta_t\lambda_t\sqrt{2\mathbf{D}_\psi(x^*, x_t)} + 16\eta_t^2\lambda_t^2}$. For $(c)$, we use the inequalities $(a+b)^2 \leq 2a^2 + 2b^2$ and $\mathbb{E}\left[(X - \mathbb{E}[X])^2\right] \leq \mathbb{E}\left[X^2\right]$. For $(d)$, we use the fact $\|\theta_t^u\|_*^2 \leq 4\lambda_t^2$ to get $\mathbb{E}\left[\|\theta_t^u\|_*^4 \mid \mathcal{F}_{t-1}\right] \leq 4\lambda_t^2\mathbb{E}\left[\|\theta_t^u\|_*^2 \mid \mathcal{F}_{t-1}\right]$. For $(e)$, we use the fact that $\|\theta_t^u\|_* \leq 2\lambda_t$ and

$$z_t\eta_t\|x^* - x_t\| \leq \frac{\eta_t\|x^* - x_t\|}{2\eta_t\lambda_t\sqrt{2\mathbf{D}_\psi(x^*, x_t)}} \leq \frac{1}{2\lambda_t}.$$

We obtain $\mathbb{E}\left[\exp\left(Z_t\right) \mid \mathcal{F}_{t-1}\right] \leq 1$. Therefore

$$\mathbb{E}\left[\exp\left(S_t\right) \mid \mathcal{F}_{t-1}\right] = \exp\left(S_{t-1}\right)\mathbb{E}\left[\exp\left(Z_t\right) \mid \mathcal{F}_{t-1}\right] \leq \exp\left(S_{t-1}\right).$$

which means $\left(\exp\left(S_t\right)\right)_{t\geq 1}$ is a supermartingale. By Ville's inequality, we have, for all $k \geq 1$

$$\Pr\left[S_k \geq \log\frac{1}{\delta}\right] \leq \delta\mathbb{E}\left[\exp\left(S_1\right)\right] \leq \delta.$$

In other words, with probability at least $1 - \delta$, for all $k \geq 1$

$$\sum_{t=1}^{k} Z_t \leq \log\frac{1}{\delta}.$$

Plugging in the definition of $Z_t$ we have

$$\sum_{t=1}^{k} z_t\eta_t\Delta_{t+1} + \sum_{t=1}^{k}\left(z_t\mathbf{D}_\psi(x^*, x_{t+1}) - z_t\mathbf{D}_\psi(x^*, x_t)\right)$$

$$\leq \log\frac{1}{\delta} + \sum_{t=1}^{k} z_t\eta_t\langle x^* - x_t, \theta_t^b\rangle + 2\sum_{t=1}^{k} z_t\eta_t^2\|\theta_t^b\|_*^2$$

$$+ \sum_{t=1}^{k} \left( \left( 2z_t \eta_t^2 + \frac{3}{8\lambda_t^2} + 24 z_t^2 \eta_t^4 \lambda_t^2 \right) \mathbb{E} \left[ \|\theta_t^u\|_*^2 \mid \mathcal{F}_{t-1} \right] \right).$$

Note that we have $z_t$ is a decreasing sequence, hence the LHS of the above inequality can be bounded by

$$\text{LHS} = \sum_{t=1}^{k} z_t \eta_t \Delta_{t+1} + z_k \mathbf{D}_\psi (x^*, x_{k+1}) - z_1 \mathbf{D}_\psi (x^*, x_1) + \sum_{t=2}^{k} (z_{k-1} - z_k) \mathbf{D}_\psi (x^*, x_k)$$

$$\geq \sum_{t=1}^{k} z_t \eta_t \Delta_{t+1} + z_k \mathbf{D}_\psi (x^*, x_{k+1}) - z_1 \mathbf{D}_\psi (x^*, x_1).$$

We obtain from here the desired inequality. $\qquad \square$

*Proof of Proposition 4.8.* We will prove by induction that on $k$

$$\sum_{i=1}^{k} \eta_i \Delta_{i+1} + \mathbf{D}_\psi (x^*, x_{k+1}) \leq \frac{1}{2} (R_1 + 8AC_1)^2.$$

The base case $k = 0$ is trivial. We have $\mathbf{D}_\psi (x^*, x_1) = \frac{R_1^2}{2}$. Suppose the statement is true for all $t \leq k \leq \ell$. Now, we show for $k + 1$. Recall that

$$z_t = \frac{1}{2\eta_t \lambda_t \max_{i \leq t} \sqrt{2\mathbf{D}_\psi (x^*, x_i)} + 16Q\eta_t^2 \lambda_t^2}.$$

Let us choose $Q = A > 1$. By the induction hypothesis, we have $\max_{i \leq t} \sqrt{2\mathbf{D}_\psi (x^*, x_i)} \leq R_1 + 8AC_1$, which implies

$$z_k \geq \frac{1}{2\eta_k \lambda_k (R_1 + 8AC_1) + 16A\eta_k^2 \lambda_k^2} = \frac{1}{2C_1 (R_1 + 16AC_1)}.$$

For an upperbound, since $\sqrt{2\mathbf{D}_\psi (x^*, x_1)} = R_1$, we have:

$$z_t \leq \frac{1}{2C_1 (R_1 + 8AC_1)}.$$

Since $z_k$ is a decreasing sequence, we have

$$z_k \sum_{t=1}^{k} \eta_t \Delta_{t+1} + z_k \mathbf{D}_\psi (x^*, x_{k+1}) \leq z_1 \mathbf{D}_\psi (x^*, x_1) + \log \frac{1}{\delta} + \sum_{t=1}^{k} z_t \eta_t \langle x^* - x_t, \theta_t^b \rangle + 2 \sum_{t=1}^{k} z_t \eta_t^2 \|\theta_t^b\|_*^2$$

$$+ \sum_{t=1}^{k} \left( \left( 2z_t \eta_t^2 + \frac{3}{8\lambda_t^2} + 24 z_t^2 \eta_t^4 \lambda_t^2 \right) \mathbb{E} \left[ \|\theta_t^u\|_*^2 \mid \mathcal{F}_{t-1} \right] \right).$$

By the choice of $\lambda_t$, for all $t \leq k$, $\|\nabla f(x_t)\|_* \leq \frac{\lambda_t}{2}$, we can apply Lemma 2.1 and have

$$\|\theta_t^b\|_* \leq 4\sigma^p \lambda_t^{1-p};$$

$$\mathbb{E} \left[ \|\theta_t^u\|_*^2 \mid \mathcal{F}_{t-1} \right] \leq 40\sigma^p \lambda_t^{2-p}.$$

Thus, we have

$$z_k \sum_{t=1}^{k} \eta_t \Delta_{t+1} + z_k \mathbf{D}_\psi (x^*, x_{k+1})$$

$$\leq z_1 \mathbf{D}_\psi (x^*, x_1) + \log \frac{1}{\delta} + 4 \sum_{t=1}^{k} z_t \eta_t \sigma^p \lambda_t^{1-p} \sqrt{2\mathbf{D}_\psi (x^*, x_t)} + 32 \sum_{t=1}^{k} z_t \eta_t^2 \sigma^{2p} \lambda_t^{2-2p}$$

$$+ 40 \sum_{t=1}^{k} \left( \left( 2z_t \eta_t^2 + \frac{3}{8\lambda_t^2} + 24 z_t^2 \eta_t^4 \lambda_t^2 \right) \sigma^p \lambda_t^{2-p} \right)$$

$$\leq z_1 \mathbf{D}_\psi\left(x^*, x_1\right) + \log\frac{1}{\delta} + \frac{2C_1\left(R_1 + 8AC_1\right)\sigma^p}{C_1\left(R_1 + 8AC_1\right)}\sum_{t=1}^{k}\left(\frac{1}{\lambda_t}\right)^p + \frac{16C_1^2\sigma^{2p}}{C_1\left(R_1 + 8AC_1\right)}\sum_{t=1}^{k}\left(\frac{1}{\lambda_t}\right)^{2p}$$

$$+ 40\left(\frac{C_1^2}{C_1\left(R_1 + 8AC_1\right)} + \frac{3}{8} + \frac{6C_1^4}{C_1^2\left(R_1 + 8AC_1\right)^2}\right)\sigma^p\sum_{t=1}^{k}\left(\frac{1}{\lambda_t}\right)^p$$

$$\leq\frac{R_1^2}{4\left(C_1R_1 + 8AC_1^2\right)} + \log\frac{1}{\delta} + 2\sigma^pC_2 + \frac{2\sigma^{2p}C_2C_3}{A} + 24\sigma^pC_2$$

$$\leq\frac{R_1^2}{4\left(C_1R_1 + 8AC_1^2\right)} + A,$$

where for the last inequality we use $\sum_{t=1}^{k}\left(\frac{1}{\lambda_t}\right)^p \leq C_2$ and $\left(\frac{1}{\lambda_t}\right)^{2p} \leq C_3\left(\frac{1}{\lambda_t}\right)^p$. We obtain

$$\sum_{t=1}^{k}\eta_t\Delta_{t+1} + \mathbf{D}_\psi\left(x^*, x_{k+1}\right) \leq 2C_1\left(R_1 + 16AC_1\right)\left(\frac{R_1^2}{4\left(C_1R_1 + 8AC_1^2\right)} + A\right)$$

$$= \frac{1}{2}R_1^2 + \frac{4AC_1^2R_1^2}{C_1R_1 + 8AC_1^2} + 2A\left(C_1R_1 + 16AC_1^2\right)$$

$$\leq \frac{1}{2}R_1^2 + 6AC_1R_1 + 32A^2C_1^2$$

$$\leq \frac{1}{2}\left(R_1 + 8AC_1\right)^2.$$

$\square$

*Proof of Theorem 4.1.* Note that our choice of $\eta$ ensures $\eta \leq \frac{R_1}{16}\frac{1}{4LR_1} \leq \frac{1}{4L}$. We have that with probability at least $1 - \delta$, event $E(\delta)$ happens. Conditioning on this event, in 4.8 we choose

$$C_1 = \frac{R_1}{24\gamma}; \quad C_2 = \frac{\gamma}{26\sigma^p}; \quad C_3 = \frac{\gamma}{26T\sigma^p}; \quad A = 3\gamma.$$

We have

$$\lambda_t\eta_t = C_1$$

$$\sum_{t=1}^{T}\left(\frac{1}{\lambda_t}\right)^p \leq \sum_{t=1}^{T}\left(\frac{\gamma}{26T}\right)\frac{1}{\sigma^p} = C_2$$

$$\left(\frac{1}{\lambda_t}\right)^{2p} \leq \frac{1}{\sigma^p}\left(\frac{\gamma}{26T}\right)\left(\frac{1}{\lambda_t}\right)^p = C_3\left(\frac{1}{\lambda_t}\right)^p$$

$$\max\left\{\log\frac{1}{\delta} + 26\sigma^pC_2 + \frac{2\sigma^{2p}C_2C_3}{A}; 1\right\} \leq 3\gamma = A.$$

We only need to show that for all $t$

$$\|\nabla f(x_t)\|_* \leq \frac{\lambda_t}{2}.$$

We will show this by induction. Indeed, we have

$$\|\nabla f(x_1)\|_* \leq \nabla_1 \leq \frac{\lambda_1}{2}.$$

Suppose that it is true for all $t \leq k$. We prove that

$$\|\nabla f(x_{k+1})\|_* \leq \frac{\lambda_{k+1}}{2}.$$

By 4.8 we have

$$\|x_{k+1} - x^*\| \leq \sqrt{2\mathbf{D}_\psi\left(x^*, x_{k+1}\right)} \leq R_1 + 8AC_1 = 2R_1.$$

Thus

$$\|\nabla f(x_{k+1})\|_* \leq \|\nabla f(x_{k+1}) - \nabla f(x^*)\|_* + \|\nabla f(x_1) - \nabla f(x^*)\|_* + \|\nabla f(x_1)\|_*$$
$$\leq L\|x_{k+1} - x^*\| + L\|x_1 - x^*\| + \nabla_1$$
$$\leq 3LR_1 + \nabla_1 \leq \frac{\lambda_{k+1}}{2}$$

as needed. Therefore from Lemma 4.7 we have

$$\eta \sum_{t=1}^{T} \Delta_{t+1} + \mathbf{D}_\psi (x^*, x_{T+1}) \leq 2R_1^2,$$

which gives

$$\frac{1}{T} \sum_{t=2}^{T+1} \Delta_t \leq \frac{2R_1^2}{\eta} = 48R_1 \max \left\{ 26^{\frac{1}{p}} T^{\frac{1-p}{p}} \sigma \gamma^{\frac{p-1}{p}}; 2\left(3LR_1 + \nabla_1\right) T^{-1} \gamma \right\}.$$

$\square$

**Theorem C.2.** *Assume that $f$ satisfies Assumption (1), (2), (3), (4) and (5). Let $\gamma = \max \left\{ \log \frac{1}{\delta}; 1 \right\}$; $R_1 = \sqrt{2\mathbf{D}_\psi (x^*, x_1)}$ assume that $\nabla_1$ is an upper bound of $\|\nabla f(x_1)\|_*$. For unknown $T$, we choose*

$$\lambda_t = \max \left\{ \left( \frac{52t\left(1 + \log t\right)^2}{\gamma} \right)^{1/p} \sigma; 2\left(3LR_1 + \nabla_1\right) \right\}, \text{ and}$$

$$\eta_t = \frac{R_1}{24\lambda_t \gamma} = \frac{R_1}{24\gamma} \min \left\{ \left( \frac{52t\left(1 + \log t\right)^2}{\gamma} \right)^{-1/p} \sigma^{-1}; \frac{1}{2}\left(3LR_1 + \nabla_1\right)^{-1} \right\}.$$

*Then with probability at least $1 - \delta$*

$$\frac{1}{T} \sum_{t=2}^{T+1} \Delta_t \leq 48R_1 \max \left\{ 52^{\frac{1}{p}} T^{\frac{1-p}{p}} \left(1 + \log T\right)^{\frac{2}{p}} \sigma \gamma^{\frac{p-1}{p}}; 2\left(3LR_1 + \nabla_1\right) T^{-1} \gamma \right\} = \widetilde{O}\left( T^{\frac{1-p}{p}} \right).$$

*Proof.* We can follow the similar steps. Notice that $(\eta_t)$ is a decreasing sequence. We also use Fact B.4 to verify the second condition of Proposition 4.8. The proof is omitted. $\square$

*Proof of Theorem 4.4.* Note that $\eta_t \leq \frac{1}{4L}$. We have that with probability at least $1 - \delta$, event $E(\delta)$ happens. Conditioning on this event, in 4.8. We choose

$$C_1 = \frac{c_1}{24}; \quad C_2 = \frac{1}{26c_2}; \quad C_3 = \frac{1}{52c_2}; \quad A = \gamma + \frac{2\sigma^p}{c_2}.$$

We verify the conditions of Proposition 4.8

$$\lambda_t \eta_t = C_1$$

$$\sum_{t=1}^{T} \left( \frac{1}{\lambda_t} \right)^p \leq \sum_{t=1}^{T} \frac{1}{52t(1 + \log t)^2 c_2} \leq \frac{1}{26c_2} = C_2$$

$$\left( \frac{1}{\lambda_t} \right)^{2p} \leq \frac{1}{52tc_2} \left( \frac{1}{\lambda_t} \right)^p \leq C_3 \left( \frac{1}{\lambda_t} \right)^p$$

$$\max \left\{ \log \frac{1}{\delta} + 26\sigma^p C_2 + \frac{2\sigma^{2p} C_2 C_3}{A}; 1 \right\} = \max \left\{ \log \frac{1}{\delta} + \frac{\sigma^p}{c_2} + \frac{\sigma^p}{c_2}; 1 \right\} \leq A,$$

where we have $\frac{2\sigma^{2p} C_2 C_3}{A} \leq 2\sigma^{2p} C_2 C_3 \times \frac{c_2}{2\sigma^p} \leq \frac{\sigma^p}{c_2}$. Also, note that

$$\|\nabla f(x_t)\|_* \leq \|\nabla f(x_t) - \nabla f(x_1)\|_* + \|\nabla f(x_1)\|_*$$

$$\leq L\|x_t - x_1\|_* + \|\nabla f(x_1)\|_* \leq \frac{\lambda_t}{2}.$$

Therefore, from Lemma 4.7, we have

$$\eta_T \sum_{t=1}^{T} \Delta_{t+1} + \mathbf{D}_\psi \left( x^*, x_{T+1} \right) \leq \frac{1}{2} \left( R_1 + 8AC_1 \right)^2$$

$$= \frac{1}{2} \left( R_1 + \frac{c_1}{3} \left( \gamma + \frac{2\sigma^p}{c_2} \right) \right)^2$$

which gives

$$\frac{1}{T} \sum_{t=2}^{T+1} \Delta_t \leq \frac{1}{2T\eta_T} \left( R_1 + \frac{c_1}{3} \left( \gamma + \frac{2\sigma^p}{c_2} \right) \right)^2$$

$$= \frac{8}{Tc_1} \left( R_1 + \frac{c_1}{3} \left( \gamma + \frac{2\sigma^p}{c_2} \right) \right)^2 \max \left\{ \left( 52T(1 + \log T)^2 c_2 \right)^{1/p} ; 2 \left( L \max_{i \leq T} \| x_i - x_1 \| + \nabla_1 \right) ; \frac{L}{8} \right\}.$$

Note that

$$\| x_i - x_1 \| \leq \| x_i - x^* \| + \| x_1 - x^* \|$$

$$\leq 2R_1 + \frac{c_1}{3} \left( \gamma + \frac{2\sigma^p}{c_2} \right)$$

which gives us the final convergence rate. $\square$

## D   Clipped Accelerated Stochastic Mirror Descent

In this section, we extend the analysis of Clipped-SMD to the case of Clipped Accelerated Stochastic Mirror Descent (Algorithm 3). We will see that the analysis is basically the same with little modification. We present in Algorithm 3 the clipped version of accelerated stochastic mirror descent (see [15]), where the clipped gradient $\widetilde{\nabla} f(x_t)$ is used to update the iterates in place of the stochastic gradient $\widehat{\nabla} f(x_t)$.

We use the following additional assumption:

**(5') Global minimizer**: We assume that $\nabla f(x^*) = 0$.

**Theorem D.1.** *Assume that $f$ satisfies Assumption (1), (2), (3), (4) and (5'). Let $\gamma = \max \left\{ \log \frac{1}{\delta}; 1 \right\}$; and $R_1 = \sqrt{2\mathbf{D}_\psi \left( x^*, x_1 \right)}$.*

*1. For known $T$, we choose a constant $c$ and $\lambda_t$ and $\eta_t$ such that*

$$c = \max \left\{ 10^4; \frac{4 \left( T + 1 \right) \left( \frac{26T}{\gamma} \right)^{\frac{1}{p}} \sigma}{\gamma L R_1} \right\},$$

$$\lambda_t = \frac{c R_1 \gamma L \alpha_t}{8} = \max \left\{ \frac{10^4 R_1 \gamma L}{6(t+1)}; \frac{T+1}{t+1} \left( \frac{26T}{\gamma} \right)^{1/p} \sigma \right\},$$

$$\eta_t = \frac{1}{3c\gamma^2 L \alpha_t} = \frac{R_1}{24\gamma} \min \left\{ \frac{4(t+1)}{10^4 R_1 \gamma L}; \frac{t+1}{T+1} \left( \frac{26T}{\gamma} \right)^{-1/p} \sigma^{-1} \right\}.$$

*Then with probability at least $1 - \delta$*

$$f \left( y_{T+1} \right) - f \left( x^* \right) \leq 6 \max \left\{ 10^4 L\gamma^2 R_1^2 (T+1)^{-2}; 4R_1 \left( T+1 \right)^{-1} (26T)^{\frac{1}{p}} \gamma^{\frac{p-1}{p}} \sigma \right\}.$$

*2. For unknown $T$, we choose $c_t$, $\lambda_t$ and $\eta_t$ such that*

$$c_t = \max \left\{ 10^4; \frac{4 \left( t + 1 \right) \left( \frac{52t(1+\log t)^2}{\gamma} \right)^{\frac{1}{p}} \sigma}{\gamma L R_1} \right\},$$

$$\lambda_t = \frac{c_t R_1 \gamma L \alpha_t}{8} = \max\left\{\frac{10^4 R_1 \gamma L}{4(t+1)}; \left(\frac{52t\left(1+\log t\right)^2}{\gamma}\right)^{1/p}\sigma\right\},$$

$$\eta_t = \frac{1}{3c_t\gamma^2 L\alpha_t} = \frac{R_1}{24\gamma}\min\left\{\frac{4(t+1)}{10^4 R_1 \gamma L}; \left(\frac{52t\left(1+\log t\right)^2}{\gamma}\right)^{-1/p}\sigma^{-1}\right\}.$$

*Then with probability at least $1-\delta$*

$$f\left(y_{T+1}\right) - f\left(x^*\right) \le 6\max\left\{10^4 L\gamma^2 R_1^2 (T+1)^{-2}; 4R_1\left(T+1\right)^{-1}\left(52T\left(1+\log T\right)^2\right)^{\frac{1}{p}}\gamma^{\frac{p-1}{p}}\sigma\right\}.$$

*Remark* D.2. One feature of the accelerated algorithm is the interpolation between the two regimes: When $\sigma$ is large, the algorithm achieves the $O\left(T^{\frac{1-p}{p}}\right)$ convergence rate, which is the same as unaccelerated algorithms; however, when $\sigma$ is sufficiently small, the algorithm achieves the accelerated $O\left(T^{-2}\right)$ rate.

We also start the analysis of accelerated stochastic mirror descent with the following lemma.

**Lemma D.3.** *Assume that $f$ satisfies Assumption (1), (2), (3), (4) and $\eta_t \le \frac{1}{2L\alpha_t}$, the iterate sequence $(x_t)_{t\ge 1}$ output by Algorithm 2 satisfies the following*

$$\frac{\eta_t}{\alpha_t}\left(f\left(y_{t+1}\right) - f\left(x^*\right)\right) - \frac{\eta_t\left(1-\alpha_t\right)}{\alpha_t}\left(f\left(y_t\right) - f\left(x^*\right)\right) + \mathbf{D}_\psi\left(x^*, z_{t+1}\right) - \mathbf{D}_\psi\left(x^*, z_t\right)$$

$$\le \eta_t\left\langle\theta_t^u, x^* - z_t\right\rangle + \eta_t\left\langle\theta_t^b, x^* - z_t\right\rangle + 2\eta_t^2\left(\left\|\theta_t^u\right\|_*^2 - \mathbb{E}\left[\left\|\theta_t^u\right\|_*^2 \mid \mathcal{F}_{t-1}\right]\right) + 2\eta_t^2\left\|\theta_t^b\right\|_*^2 + 2\eta_t^2\mathbb{E}\left[\left\|\theta_t^u\right\|_*^2 \mid \mathcal{F}_{t-1}\right].$$

*Proof of Lemma D.3.* We have

$$f\left(y_{t+1}\right) - f\left(x^*\right) = \underbrace{f\left(y_{t+1}\right) - f\left(x_t\right)}_{\text{smoothness}} + \underbrace{f\left(x_t\right) - f\left(x^*\right)}_{\text{convexity}}$$

$$\le \left\langle\nabla f\left(x_t\right), y_{t+1} - x_t\right\rangle + \frac{L}{2}\left\|y_{t+1} - x_t\right\|^2$$

$$+ \alpha_t\left\langle\nabla f\left(x_t\right), x_t - x^*\right\rangle + \left(1-\alpha_t\right)\left(f\left(x_t\right) - f\left(x^*\right)\right)$$

$$= \underbrace{\left(1-\alpha_t\right)\left\langle\nabla f\left(x_t\right), y_t - x_t\right\rangle}_{\text{convexity}} + \alpha_t\left\langle\nabla f\left(x_t\right), z_{t+1} - x^*\right\rangle$$

$$+ \frac{L\alpha_t^2}{2}\left\|z_{t+1} - z_t\right\|^2 + \left(1-\alpha_t\right)\left(f\left(x_t\right) - f\left(x^*\right)\right)$$

$$\le \left(1-\alpha_t\right)\left(f\left(y_t\right) - f\left(x_t\right)\right) + \left(1-\alpha_t\right)\left(f\left(x_t\right) - f\left(x^*\right)\right)$$

$$+ \alpha_t\left\langle\theta_t, x^* - z_{t+1}\right\rangle + \alpha_t\left\langle\widetilde{\nabla} f(x_t), z_{t+1} - x^*\right\rangle + \frac{L\alpha_t^2}{2}\left\|z_{t+1} - z_t\right\|^2$$

$$\le \left(1-\alpha_t\right)\left(f\left(y_t\right) - f\left(x^*\right)\right) + \alpha_t\left\langle\theta_t, x^* - z_{t+1}\right\rangle$$

$$+ \alpha_t\left\langle\widetilde{\nabla} f(x_t), z_{t+1} - x^*\right\rangle + \frac{L\alpha_t^2}{2}\left\|z_{t+1} - z_t\right\|^2.$$

By the optimality condition, we have

$$\left\langle\eta_t\widetilde{\nabla} f(x_t) + \nabla_x\mathbf{D}_\psi\left(z_{t+1}, z_t\right), x^* - z_{t+1}\right\rangle \ge 0$$

and thus

$$\left\langle\eta_t\widetilde{\nabla} f(x_t), z_{t+1} - x^*\right\rangle \le \left\langle\nabla_x\mathbf{D}_\psi\left(z_{t+1}, z_t\right), x^* - z_{t+1}\right\rangle.$$

Note that

$$\left\langle\nabla_x\mathbf{D}_\psi\left(z_{t+1}, z_t\right), x^* - z_{t+1}\right\rangle = \left\langle\nabla\psi\left(z_{t+1}\right) - \nabla\psi\left(z_t\right), x^* - z_{t+1}\right\rangle$$

$$= \mathbf{D}_\psi\left(x^*, z_t\right) - \mathbf{D}_\psi\left(z_{t+1}, z_t\right) - \mathbf{D}_\psi\left(x^*, z_{t+1}\right).$$

Thus

$$\eta_t \left\langle \widetilde{\nabla} f(x_t), z_{t+1} - x^* \right\rangle \leq \mathbf{D}_\psi \left( x^*, z_t \right) - \mathbf{D}_\psi \left( x^*, z_{t+1} \right) - \mathbf{D}_\psi \left( z_{t+1}, z_t \right)$$

$$\leq \mathbf{D}_\psi \left( x^*, z_t \right) - \mathbf{D}_\psi \left( x^*, z_{t+1} \right) - \frac{1}{2} \left\| z_{t+1} - z_t \right\|^2$$

where we have used that $\mathbf{D}_\psi \left( z_{t+1}, z_t \right) \geq \frac{1}{2} \left\| z_{t+1} - z_t \right\|^2$ by the strong convexity of $\psi$. We have

$$f \left( y_{t+1} \right) - f \left( x^* \right) \leq \left( 1 - \alpha_t \right) \left( f \left( y_t \right) - f \left( x^* \right) \right) + \alpha_t \left\langle \theta_t, x^* - z_{t+1} \right\rangle$$

$$+ \frac{\alpha_t}{\eta_t} \mathbf{D}_\psi \left( x^*, z_t \right) - \frac{\alpha_t}{\eta_t} \mathbf{D}_\psi \left( x^*, z_{t+1} \right) + \left( \frac{L\alpha_t^2}{2} - \frac{\alpha_t}{2\eta_t} \right) \left\| z_{t+1} - z_t \right\|^2 .$$

Dividing both sides by $\frac{\alpha_t}{\eta_t}$ and using the condition $L\eta_t \alpha_t \leq \frac{1}{2}$, we have

$$\frac{\eta_t}{\alpha_t} \left( f \left( y_{t+1} \right) - f \left( x^* \right) \right) + \mathbf{D}_\psi \left( x^*, z_{t+1} \right) - \mathbf{D}_\psi \left( x^*, z_t \right)$$

$$\leq \frac{\eta_t \left( 1 - \alpha_t \right)}{\alpha_t} \left( f \left( y_t \right) - f \left( x^* \right) \right) + \eta_t \left\langle \theta_t, x^* - z_t \right\rangle$$

$$+ \eta_t \left\langle \theta_t, z_t - z_{t+1} \right\rangle - \frac{1 - L\eta_t \alpha_t}{2} \left\| z_{t+1} - z_t \right\|^2$$

$$\leq \frac{\eta_t \left( 1 - \alpha_t \right)}{\alpha_t} \left( f \left( y_t \right) - f \left( x^* \right) \right) + \eta_t \left\langle \theta_t, x^* - z_t \right\rangle$$

$$+ \frac{\eta_t^2 \left\| \theta_t \right\|_*^2}{2 \left( 1 - L\eta_t \alpha_t \right)}$$

$$\leq \frac{\eta_t \left( 1 - \alpha_t \right)}{\alpha_t} \left( f \left( y_t \right) - f \left( x^* \right) \right) + \eta_t \left\langle \theta_t^u + \theta_t^b, x^* - z_t \right\rangle$$

$$+ 2\eta_t^2 \left\| \theta_t^u \right\|_*^2 + 2\eta_t^2 \left\| \theta_t^b \right\|_*^2$$

as needed. $\qquad\square$

Similarly to the previous section, we define the following variables

$$Z_t = z_t \left( \frac{\eta_t}{\alpha_t} \left( f \left( y_{t+1} \right) - f \left( x^* \right) \right) - \frac{\eta_t \left( 1 - \alpha_t \right)}{\alpha_t} \left( f \left( y_t \right) - f \left( x^* \right) \right) + \mathbf{D}_\psi \left( x^*, z_{t+1} \right) - \mathbf{D}_\psi \left( x^*, z_t \right) \right.$$

$$\left. - \eta_t \left\langle \theta_t^b, x^* - z_t \right\rangle - 2\eta_t^2 \left\| \theta_t^b \right\|_*^2 - 2\eta_t^2 \mathbb{E} \left[ \left\| \theta_t^u \right\|_*^2 \mid \mathcal{F}_{t-1} \right] \right) - \left( \frac{3}{8\lambda_t^2} + 24 z_t^2 \eta_t^4 \lambda_t^2 \right) \mathbb{E} \left[ \left\| \theta_t^u \right\|^2 \mid \mathcal{F}_{t-1} \right],$$

where $z_t = \dfrac{1}{2\eta_t \lambda_t \max_{i \leq t} \sqrt{2\mathbf{D}_\psi \left( x^*, x_i \right) + 16Q\eta_t^2 \lambda_t^2}}$

for a constant $Q \geq 1$. We also let $S_t = \sum_{i=1}^t Z_i$. Following the same analysis as in previous sections, we can obtain Lemma D.4 and Proposition D.5, for which we will omit the proofs here. The only step we need to pay attention to when showing Lemma D.4 is when we bound the sum

$$\sum_{t=1}^k \frac{z_t \eta_t}{\alpha_t} \left( f \left( y_{t+1} \right) - f \left( x^* \right) \right) - \frac{z_t \eta_t \left( 1 - \alpha_t \right)}{\alpha_t} \left( f \left( y_t \right) - f \left( x^* \right) \right).$$

If we assume $\frac{\eta_{t-1}}{\alpha_{t-1}} \geq \frac{\eta_t \left( 1 - \alpha_t \right)}{\alpha_t}$, since $z_t$ is a decreasing sequence and $\alpha_1 = 0$, we can lower bound the above sum by the last term $\frac{z_k \eta_k}{\alpha_k} \left( f \left( y_{k+1} \right) - f \left( x^* \right) \right)$, which gives us the desired inequality.

**Lemma D.4.** *Assume that for all $t \geq 1$, $\eta_t$ satisfies $\frac{\eta_{t-1}}{\alpha_{t-1}} \geq \frac{\eta_t \left( 1 - \alpha_t \right)}{\alpha_t}$. For any $\delta > 0$, let $E(\delta)$ be the event that for all $1 \leq k \leq T$*

$$\frac{z_k \eta_k}{\alpha_k} \left( f \left( y_{k+1} \right) - f \left( x^* \right) \right) + z_k \mathbf{D}_\psi \left( x^*, x_{k+1} \right)$$

$$\leq z_1 \mathbf{D}_\psi\left(x^*, x_1\right) + \log\frac{1}{\delta} + \sum_{t=1}^{k} z_t \eta_t \left\langle x^* - x_t, \theta_t^b \right\rangle + 2\sum_{t=1}^{k} z_t \eta_t^2 \left\|\theta_t^b\right\|_*^2$$

$$+ \sum_{t=1}^{k}\left(\left(2z_t\eta_t^2 + \frac{3}{8\lambda_t^2} + 24z_t^2\eta_t^4\lambda_t^2\right) \mathbb{E}\left[\left\|\theta_t^u\right\|_*^2 \mid \mathcal{F}_{t-1}\right]\right).$$

*Then* $\Pr\left[E(\delta)\right] \geq 1 - \delta$.

Finally, we state a general condition for the choice of $\eta_t$ and $\lambda_t$, which follows exactly the same as in Proposition 4.8. The proof for Theorem D.1 is a direct consequence of this.

**Proposition D.5.** *We assume that the event $E(\delta)$ from Lemma D.4 happens. Suppose that for some $\ell \leq T$, there are constants $C_1$ and $C_2$ such that for all $t \leq \ell$*

*1.* $\lambda_t\eta_t = C_1$; *2.* $\sum_{t=1}^{\ell}\left(\frac{1}{\lambda_t}\right)^p \leq C_2$; *3.* $\left(\frac{1}{\lambda_t}\right)^{2p} \leq C_3\left(\frac{1}{\lambda_t}\right)^p$; *4.* $\|\nabla f(x_t)\|_* \leq \frac{\lambda_t}{2}$.

*Then for all* $t \leq \ell + 1$

$$\frac{\eta_t}{\alpha_t}\left(f\left(y_{t+1}\right) - f\left(x^*\right)\right) + \mathbf{D}_\psi\left(x^*, z_{t+1}\right) \leq \frac{1}{2}\left(R_1 + 8AC_1\right)^2$$

*for* $A \geq \max\left\{\log\frac{1}{\delta} + 26\sigma^p C_2 + \frac{2\sigma^{2p}C_2C_3}{A}; 1\right\}$.

*Proof of Theorem D.1.* 1. Note that $\eta_t \leq \frac{1}{2c\gamma^2 L\alpha_t} \leq \frac{1}{2L\alpha_t}$ and

$$\frac{\eta_{t-1}}{\alpha_{t-1}} = \frac{t^2}{8c\gamma^2 L}$$

$$\frac{\eta_t\left(1 - \alpha_t\right)}{\alpha_t} = \frac{(t+1)(t-1)}{8c\gamma^2 L}$$

thus $\frac{\eta_{t-1}}{\alpha_{t-1}} \geq \frac{\eta_t(1-\alpha_t)}{\alpha_t}$. We have that with probability at least $1 - \delta$, event $E(\delta)$ happens. Conditioning on this event, in 4.8 We choose

$$C_1 = \frac{R_1}{24\gamma}; \quad C_2 = \frac{\gamma}{26\sigma^p}; \quad C_3 = \frac{\gamma}{26T\sigma^p}; \quad A = 3\gamma.$$

We can verify the conditions of Proposition D.5 similarly as in previous section for these choices of $C_1, C_2$, and $C_3$.

We will show by induction that for all $t \geq 1$, $\|\nabla f(x_t)\|_* \leq \frac{\lambda_t}{2}$ and $\max\left\{\|x_t - x^*\|, \|y_t - x^*\|, \|z_t - x^*\|\right\} \leq 2R_1$.

For $t = 1$, notice that $x_1 = y_1 = z_1$. Thus, we have

$$\|\nabla f(x_1)\|_* = \|\nabla f(x_1) - \nabla f(x^*)\|_* \leq LR_1 \leq \frac{\lambda_1}{2}.$$

Now assume that the claim holds for $1 \leq t \leq k$. By Proposition D.5, we know that

$$\frac{2\eta_k}{\alpha_k}f\left(y_{k+1}\right) - f\left(x^*\right) + \|z_{k+1} - x^*\|^2 \leq 4R_1^2.$$

Furthermore

$$\|y_{k+1} - x^*\| \leq (1 - \alpha_k)\|y_k - x^*\| + \alpha_k\|z_{k+1} - x^*\| \leq 2R_1$$
$$\|x_{k+1} - x^*\| \leq (1 - \alpha_k)\|y_{k+1} - x^*\| + \alpha_k\|z_{k+1} - x^*\| \leq 2R_1$$

For $k \geq 1$ we have $\alpha_{k+1} = \frac{2}{k+2} < 1$; $\frac{\alpha_{k+1}}{1-\alpha_{k+1}} = \frac{2}{k} \leq \frac{4}{k+2} \leq 2\alpha_{t+1}$ and $\alpha_t \leq \frac{3}{2}\alpha_{t+1}$. Hence,

$$\|\nabla f(x_{k+1})\|_* \leq \|\nabla f(x_{k+1}) - \nabla f(y_{k+1})\|_* + \|\nabla f(y_{k+1}) - \nabla f(x^*)\|_*$$
$$\leq L\|x_{k+1} - y_{k+1}\| + \sqrt{2L\left(f\left(y_{k+1}\right) - f\left(x^*\right)\right)}$$
$$\leq \frac{L\alpha_{k+1}\|x_{k+1} - z_{k+1}\|}{1 - \alpha_{k+1}} + 2R_1\sqrt{\frac{L\alpha_t}{2\eta_t}}$$

$$\leq 4LR_1\frac{\alpha_{k+1}}{1-\alpha_{k+1}} + 2\sqrt{\frac{3}{2}}c\gamma R_1 L\alpha_t$$

$$\leq 8\gamma LR_1\alpha_{t+1} + 3\sqrt{\frac{3}{2}}c\gamma LR_1\alpha_{t+1}$$

$$\leq (8+3\sqrt{\frac{3}{2}}c)R_1\gamma L\alpha_{t+1}$$

$$= \frac{16(8+3\sqrt{\frac{3}{2}}c)\lambda_{t+1}}{2c} \leq \frac{\lambda_{t+1}}{2}$$

as needed. Therefore, we have

$$\frac{\eta_T}{\alpha_T}\left(f\left(y_{T+1}\right) - f\left(x^*\right)\right) + \mathbf{D}_\psi\left(x^*, x_{T+1}\right) \leq 2R_1^2$$

which gives

$$f\left(y_{T+1}\right) - f\left(x^*\right) \leq \frac{2R_1^2\alpha_T}{\eta_T} = 6R_1^2c\gamma^2 L\alpha_T^2$$

$$= 6\max\left\{10^4L\gamma^2R_1^2(T+1)^{-2}; 6R_1\left(T+1\right)^{-1}\left(26T\right)^{\frac{1}{p}}\gamma^{\frac{p-1}{p}}\sigma\right\}.$$

2. Following the similar steps to the proof of Theorem D.1, and noticing that $(c_t)$ is a increasing sequence, we obtain the convergence rate. $\qquad\square$

