167 *for $0 \le \lambda \le \frac{1}{R}$*

$$\mathbb{E}[\exp(\lambda X)] \le \exp\left( \frac{3}{4} \lambda^2 \mathbb{E}[X^2] \right).$$

## 3 Clipped Stochastic Gradient Descent for Nonconvex Functions

169 In this section, we study the convergence of Clipped-SGD for nonconvex functions. Here, we con-
170 sider the domain to be $\mathbb{R}^d$ equipped with the standard $\ell_2$ norm. We first outline a blackbox concen-
171 tration argument to show convergence in high probability of Algorithm 1 and then follow-up with a
172 more powerful whitebox approach that allows for a tight high probability convergence analysis.

**Algorithm 1** Clipped-SGD

---
Parameters: initial point $x_1$, step sizes $\{\eta_t\}$, clipping parameters $\{\lambda_t\}$
for $t = 1$ to $T$ do
$$\widetilde{\nabla} f(x_t) = \min\left\{1, \frac{\lambda_t}{\|\widehat{\nabla} f(x_t)\|}\right\} \widehat{\nabla} f(x_t)$$
$$x_{t+1} = x_t - \eta_t \widetilde{\nabla} f(x_t)$$

---

**Comparison to previous works.** In the simple setting of known time horizon and without momentum for Clipped-SGD, the $\widetilde{O}(T^{\frac{2-2p}{3p-2}})$ convergence rate has not been shown before to the best of our knowledge. The recent work by [27] study this case and only give a suboptimal rate of $\widetilde{O}(T^{\frac{1-p}{p}})$. Note that [1, 18] study other variants of Clipped-SGD with momentums incorporated. Although [1, 18] achieve the nearly-optimal time dependency of $\widetilde{O}(T^{\frac{2-2p}{3p-2}})$ in the non-convex settings, they rely on using blackbox concentration inequalities which result in a suboptimal convergence rate that also requires a known time horizon.

We first present the guarantee for known time horizon $T$ via our whitebox approach in Theorem 3.1 and defer the statement for unknown $T$ in Theorem B.2 to the appendix.

**Theorem 3.1.** *Assume that $f$ satisfies Assumption (1'), (2), (3), (4). Let $\gamma := \max\left\{\log\frac{1}{\delta}; 1\right\}$ and $\Delta_1 := f(x_1) - f^*$. For known time horizon $T$, we choose $\lambda_t$ and $\eta_t$ such that*

$$\lambda_t := \lambda := \max\left\{\left(\frac{8\gamma}{\sqrt{L\Delta_1}}\right)^{\frac{1}{p-1}} T^{\frac{1}{3p-2}} \sigma^{\frac{p}{p-1}}; 2\sqrt{90L\Delta_1}; 32^{\frac{1}{p}} \sigma T^{\frac{1}{3p-2}}\right\}$$

$$\eta_t := \eta := \frac{\sqrt{\Delta_1} T^{\frac{1-p}{3p-2}}}{8\lambda\sqrt{L}\gamma} = \frac{\sqrt{\Delta_1}}{8\sqrt{L}\gamma} \min\left\{\left(\frac{8\gamma}{\sqrt{L\Delta_1}}\right)^{\frac{-1}{p-1}} T^{\frac{-p}{3p-2}} \sigma^{\frac{-p}{p-1}}; \frac{T^{\frac{1-p}{3p-2}}}{2\sqrt{90L\Delta_1}}; \frac{T^{\frac{-p}{3p-2}}}{32^{1/p}\sigma}\right\}.$$

*Then with probability at least $1 - \delta$*

$$\frac{1}{T} \sum_{t=1}^{T} \|\nabla f(x_t)\|^2 \leq 720\sqrt{\Delta_1 L}\gamma \max\left\{\left(\frac{8\gamma}{\sqrt{L\Delta_1}}\right)^{\frac{1}{p-1}} T^{\frac{2-2p}{3p-2}} \sigma^{\frac{p}{p-1}};\right.$$

$$\left. 2\sqrt{90L\Delta_1} T^{\frac{1-2p}{3p-2}}; 32^{1/p} \sigma T^{\frac{2-2p}{3p-2}}\right\} = O\left(T^{\frac{2-2p}{3p-2}}\right).$$

Lemma 3.2 is key and provides the starting point of the analysis. Its proof is shown in the Appendix.

**Lemma 3.2.** *Assume that $f$ satisfies Assumption (1'), (2), (3), (4) and $\eta_t \leq \frac{1}{L}$ then for all $t \geq 1$,*

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

 ## C   Missing Proofs from Section 4

**Lemma C.1.** *Suppose that $\eta_t \leq \frac{1}{4L}$ and assume $f$ satisfies Assumption (1), (2), (3) as well as the following condition*

$$f(y) - f(x) \leq \langle \nabla f(x), y - x \rangle + G \|y - x\| + \frac{L}{2} \|y - x\|^2, \quad \forall y, x \in \mathcal{X}. \tag{7}$$

*Then the iterate sequence $(x_t)_{t \geq 1}$ output by Algorithm 2 satisfies the following:*

$$\eta_t \Delta_{t+1} \leq \mathbf{D}_\psi(x^*, x_t) - \mathbf{D}_\psi(x^*, x_{t+1}) + \eta_t \langle x^* - x_t, \theta_t^u \rangle + \eta_t \langle x^* - x_t, \theta_t^b \rangle$$
$$+ 2\eta_t^2 \left( \|\theta_t^u\|_*^2 - \mathbb{E}\left[ \|\theta_t^u\|_*^2 \mid \mathcal{F}_{t-1} \right] \right) + 2\eta_t^2 \mathbb{E}\left[ \|\theta_t^u\|_*^2 \mid \mathcal{F}_{t-1} \right] + 2\eta_t^2 \|\theta_t^b\|_*^2 + 2G^2 \eta_t^2.$$

*Proof.* By condition (7) and convexity,

$$f(x_{t+1}) - f(x^*) \leq \underbrace{f(x_{t+1}) - f(x_t)}_{\text{condition (7)}} + \underbrace{f(x_t) - f(x^*)}_{\text{convexity}}$$

$$\leq \langle \nabla f(x_t), x_{t+1} - x_t \rangle + \frac{L}{2} \|x_t - x_{t+1}\|^2 + G \|x_t - x_{t+1}\| + \langle \nabla f(x_t), x_t - x^* \rangle$$

$$= \langle \nabla f(x_t), x_{t+1} - x^* \rangle + \frac{L}{2} \|x_t - x_{t+1}\|^2 + G \|x_t - x_{t+1}\|$$

$$= \langle \theta_t, x^* - x_{t+1} \rangle + \left\langle \widetilde{\nabla} f(x_t), x_{t+1} - x^* \right\rangle + \frac{L}{2} \|x_t - x_{t+1}\|^2 + G \|x_t - x_{t+1}\|.$$

By the optimality condition, we have

$$\left\langle \eta_t \widetilde{\nabla} f(x_t) + \nabla_x \mathbf{D}_\psi(x_{t+1}, x_t), x^* - x_{t+1} \right\rangle \geq 0$$

and thus

$$\left\langle \eta_t \widetilde{\nabla} f(x_t), x_{t+1} - x^* \right\rangle \leq \langle \nabla_x \mathbf{D}_\psi(x_{t+1}, x_t), x^* - x_{t+1} \rangle.$$

Note that

$$\langle \nabla_x \mathbf{D}_\psi(x_{t+1}, x_t), x^* - x_{t+1} \rangle = \langle \nabla \psi(x_{t+1}) - \nabla \psi(x_t), x^* - x_{t+1} \rangle$$

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

$$\leq \frac{1}{2} R_1^2 + 6AC_1 R_1 + 32A^2 C_1^2$$

$$\leq \frac{1}{2} \left( R_1 + 8AC_1 \right)^2.$$

$\square$

*Proof of Theorem 4.1.* Note that our choice of $\eta$ ensures $\eta \leq \frac{R_1}{16} \frac{1}{4LR_1} \leq \frac{1}{4L}$. We have that with probability at least $1 - \delta$, event $E(\delta)$ happens. Conditioning on this event, in 4.8 we choose

$$C_1 = \frac{R_1}{24\gamma}; \quad C_2 = \frac{\gamma}{26\sigma^p}; \quad C_3 = \frac{\gamma}{26T\sigma^p}; \quad A = 3\gamma.$$

We have

$$\lambda_t \eta_t = C_1$$

$$\sum_{t=1}^{T} \left( \frac{1}{\lambda_t} \right)^p \leq \sum_{t=1}^{T} \left( \frac{\gamma}{26T} \right) \frac{1}{\sigma^p} = C_2$$

$$\left( \frac{1}{\lambda_t} \right)^{2p} \leq \frac{1}{\sigma^p} \left( \frac{\gamma}{26T} \right) \left( \frac{1}{\lambda_t} \right)^p = C_3 \left( \frac{1}{\lambda_t} \right)^p$$

$$\max \left\{ \log \frac{1}{\delta} + 26\sigma^p C_2 + \frac{2\sigma^{2p} C_2 C_3}{A}; 1 \right\} \leq 3\gamma = A.$$

We only need to show that for all $t$

$$\| \nabla f(x_t) \|_* \leq \frac{\lambda_t}{2}.$$

We will show this by induction. Indeed, we have

$$\| \nabla f(x_1) \|_* \leq \nabla_1 \leq \frac{\lambda_1}{2}.$$

Suppose that it is true for all $t \leq k$. We prove that

$$\| \nabla f(x_{k+1}) \|_* \leq \frac{\lambda_{k+1}}{2}.$$

By 4.8 we have

$$\| x_{k+1} - x^* \| \leq \sqrt{2 \mathbf{D}_\psi \left( x^*, x_{k+1} \right)} \leq R_1 + 8AC_1 = 2R_1.$$

Thus

$$\| \nabla f(x_{k+1}) \|_* \leq \| \nabla f(x_{k+1}) - \nabla f(x^*) \|_* + \| \nabla f(x_1) - \nabla f(x^*) \|_* + \| \nabla f(x_1) \|_*$$

$$\leq L \left\| x_{k+1} - x^* \right\| + L \left\| x_1 - x^* \right\| + \nabla_1$$

$$\leq 3LR_1 + \nabla_1 \leq \frac{\lambda_{k+1}}{2}$$

as needed. Therefore from Lemma 4.7 we have

$$\eta \sum_{t=1}^{T} \Delta_{t+1} + \mathbf{D}_\psi \left( x^*, x_{T+1} \right) \leq 2R_1^2,$$

which gives

$$\frac{1}{T} \sum_{t=2}^{T+1} \Delta_t \leq \frac{2R_1^2}{\eta} = 48R_1 \max \left\{ 26^{\frac{1}{p}} T^{\frac{1-p}{p}} \sigma \gamma^{\frac{p-1}{p}}; 2 \left( 3LR_1 + \nabla_1 \right) T^{-1} \gamma \right\}.$$

$\square$

**Theorem C.2.** *Assume that $f$ satisfies Assumption (1), (2), (3), (4) and (5). Let $\gamma = \max \left\{ \log \frac{1}{\delta}; 1 \right\}$; $R_1 = \sqrt{2\mathbf{D}_\psi \left( x^*, x_1 \right)}$ assume that $\nabla_1$ is an upper bound of $\left\| \nabla f(x_1) \right\|_*$. For unknown $T$, we choose*

$$\lambda_t = \max \left\{ \left( \frac{52t \left( 1 + \log t \right)^2}{\gamma} \right)^{1/p} \sigma; 2 \left( 3LR_1 + \nabla_1 \right) \right\}, \; and$$

$$\eta_t = \frac{R_1}{24\lambda_t \gamma} = \frac{R_1}{24\gamma} \min \left\{ \left( \frac{52t \left( 1 + \log t \right)^2}{\gamma} \right)^{-1/p} \sigma^{-1}; \frac{1}{2} \left( 3LR_1 + \nabla_1 \right)^{-1} \right\}.$$

*Then with probability at least $1 - \delta$*

$$\frac{1}{T} \sum_{t=2}^{T+1} \Delta_t \leq 48R_1 \max \left\{ 52^{\frac{1}{p}} T^{\frac{1-p}{p}} \left( 1 + \log T \right)^{\frac{2}{p}} \sigma \gamma^{\frac{p-1}{p}}; 2 \left( 3LR_1 + \nabla_1 \right) T^{-1} \gamma \right\} = \widetilde{O} \left( T^{\frac{1-p}{p}} \right).$$

*Proof.* We can follow the similar steps. Notice that $(\eta_t)$ is a decreasing sequence. We also use Fact B.3 to verify the second condition of Proposition 4.8. The proof is omitted. $\square$

*Proof of Theorem 4.4.* Note that $\eta_t \leq \frac{1}{4L}$. We have that with probability at least $1 - \delta$, event $E(\delta)$ happens. Conditioning on this event, in 4.8. We choose

$$C_1 = \frac{c_1}{24}; \quad C_2 = \frac{1}{26c_2}; \quad C_3 = \frac{1}{52c_2}; \quad A = \gamma + \frac{2\sigma^p}{c_2}.$$

We verify the conditions of Proposition 4.8

$$\lambda_t \eta_t = C_1$$

$$\sum_{t=1}^{T} \left( \frac{1}{\lambda_t} \right)^p \leq \sum_{t=1}^{T} \frac{1}{52t(1 + \log t)^2 c_2} \leq \frac{1}{26c_2} = C_2$$

$$\left( \frac{1}{\lambda_t} \right)^{2p} \leq \frac{1}{52tc_2} \left( \frac{1}{\lambda_t} \right)^p \leq C_3 \left( \frac{1}{\lambda_t} \right)^p$$

$$\max \left\{ \log \frac{1}{\delta} + 26\sigma^p C_2 + \frac{2\sigma^{2p} C_2 C_3}{A}; 1 \right\} = \max \left\{ \log \frac{1}{\delta} + \frac{\sigma^p}{c_2} + \frac{\sigma^p}{c_2}; 1 \right\} \leq A,$$

where we have $\frac{2\sigma^{2p} C_2 C_3}{A} \leq 2\sigma^{2p} C_2 C_3 \times \frac{c_2}{2\sigma^p} \leq \frac{\sigma^p}{c_2}$. Also, note that

$$\left\| \nabla f(x_t) \right\|_* \leq \left\| \nabla f(x_t) - \nabla f(x_1) \right\|_* + \left\| \nabla f(x_1) \right\|_*$$