# OpenReview forum: "Improved Convergence in High Probability of Clipped Gradient Methods with Heavy Tailed Noise"
_NeurIPS.cc/2023/Conference — NeurIPS 2023 spotlight_

### Official Review · Reviewer_bwSN · 2023-07-03

**Soundness:** 3 good
**Presentation:** 4 excellent
**Contribution:** 3 good
**Rating:** 7
**Confidence:** 3

**Summary:**

The author(s) proposed an improved convergence analysis of clipped gradient descent with heavey tail noise. By the improved analysis, the author(s) can either improved the logarithmic dependency of $T$ or relax the known time horizon assumption.

**Strengths:**

- The author(s) proposed a new analysis framework for the high-probability error bound of clipped SGD with heavy-tail noise. The analysis is different from the classic Freedman concentration analysis, and can be used in more flexible settings compared to previous analysis, e.g., unknown time horizon, the new analysis also results in slightly better dependency of $T$.
- The writing is clear and easy to follow.
- Related works are well-addressed to my knowledge.
- Overall, I think this paper makes a decent contribution to the field if all other reviewers believe the proof is correct.

**Weaknesses:**

- It is better to include a table to compare with prior works in terms of the convegence rate and the assumptions being used. So readers can better understand the position of this work in the literature.

**Questions:**

- Ling 190, there is no $ \langle \nabla f(x_t), \theta_t \rangle $ in equation (5), is this a typo?

**Limitations:**

I do not find any nagative social impact of this work.

---

> ### Author Rebuttal · Authors · 2023-08-09
>
> We thank the reviewer for the feedback. We will add a table in the revision of the paper.
>
> In Line 190: The term $\left\langle \nabla f(x_{t}),\theta_{t}\right\rangle$  does not appear in Eq (5) because we have decompose it into $\left\langle \nabla f(x_{t}),\theta_{t}^{u}\right\rangle +\left\langle \nabla f(x_{t}),\theta_{t}^{b}\right\rangle$. We next explain how treating the term $\left\langle \nabla f(x_{t}),\theta_{t}^{b}\right\rangle$ more carefully can give us the optimal rate.

---

> > ### Comment · Reviewer_bwSN · 2023-08-17
> > **Thanks for your response**
> >
> > Thanks for your response, I am keep my score unchanged.

---

### Official Review · Reviewer_2mhk · 2023-07-06

**Soundness:** 3 good
**Presentation:** 3 good
**Contribution:** 3 good
**Rating:** 6
**Confidence:** 4

**Summary:**

The paper addresses the problem of deriving high-probability complexity bounds for the convex and non-convex optimization problems on closed convex sets under the assumption that the noise in the stochastic gradient has bounded $p$th moment for some $1<p\leq 2$. The key feature of the results is that they have improved dependence on the failure probability $\delta$ in comparison to the prior works. In the convex case, the authors also consider non-Euclidean norms, Clipped Stochastic Mirror Descent, and its accelerated version. Overall, the results are good and very relevant for the community.

**Strengths:**

1. **Improved dependence on $\delta$ in the convex case.** Previous works addressing similar setups provide complexity results that depend on $\delta$ as $\log(1/(\varepsilon\delta))$ while the rates derived in this work (in the convex case, for known horizon) for Clipped-SMD and Clipped-ASMD imply the complexities that are proportional to $\log(1/\delta)$. Although this affects only the logarithmic term, I believe it is an important contribution to the stochastic optimization literature. In particular, this requires applying the proof technique (without induction argument) that differs from the existing approaches.

2. **Constrained case for Clipped-SGD and non-Euclidean prox-structure.** This works provides high-probability convergence analysis for the constrained problems for the first time without the bounded variance assumption. Moreover, it provides an extension to the non-Euclidean case. This is an important step forward.

3. **Horizon-independent results.** The authors also provide results that are independent of the time horizon $T$, while the previous works explicitly use $T$ to choose the parameters of the methods.

4. The paper is well-written. I did not find any serious issues in the proofs.


**Weaknesses:**

1. **Deterministic term and logarithms in the results for the non-convex case.** Although the dominating terms (that contain $\sigma$) has optimal dependence on $T$, the deterministic term (the one that does not depend on $\sigma$) is $O(T^{\frac{1-2p}{3p-2}})$ that is not optimal for the deterministic case: it becomes worse when $p \to 1$, while should be $O(T^{-1})$ as for Gradient Descent.  Such a $O(T^{-1})$ deterministic term can be achieved (e.g., see Sadiev et al. (2023)). Next, the improvement of the logarithmic factor is not that evident: while under the logarithm, there is no $T$ anymore, the power of the logarithm increases. In particular, the first term in the rate from Theorem 3.1 is proportional to $\gamma^{\frac{p}{p-1}} \geq \gamma^2$. If one wants to achieve $\frac{1}{T}\sum_{t=1}^T \|\nabla f(x_t)\|^2 \leq \varepsilon$, then according to Theorem 3.1, the complexity bound will have a term proportional to $O(\frac{\log^{\frac{p(3p-2)}{2p-2}}\frac{1}{\delta}}{\varepsilon^{\frac{3p-2}{2p-2}}})$. In the best case (when $p=2$), this equals to $O(\frac{\log^{4}\frac{1}{\delta}}{\varepsilon^2})$. The corresponding term from (Sadiev et al., 2023) is $O(\frac{\log\frac{1}{\delta\varepsilon^2}}{\varepsilon^2})$. When both $\delta$ and $\varepsilon$ are small, $\log\frac{1}{\delta\varepsilon^2}$ can be much smaller than $\log^{4}\frac{1}{\delta}$. I believe this should be discussed in the paper, and a fair comparison should be provided.

2. **Logarithms in the horizon-independent results.** When $T$ is unknown, the results become worse by a polynomial factor of $\log T$ that also spoils the complexity even in the convex case (and worse than for the case of known $T$ in the prior works).

3. **Some parts of the proofs are not finished.** The proof of Lemma 3.4 is not finished. Next, although the result is believable, the proof of Theorem B.2 is not complete as well. The authors should at least provide a more detailed sketch explicitly pointing to the places that will be changed and how.

4. **Results for the accelerated method requires $\nabla f(x^\ast) = 0$.** When the gradient at the solution equals zero, then the problem becomes almost an unconstrained one. (in terms of the analysis). The authors should indicate in the main text that the accelerated algorithm is analyzed under this additional assumption.

**Questions:**

1. Line 39, "where the convergence rate depends on...": the work [32] does not have high-probability results. Do you mean that one can get such results from the results of [32] using Markov's inequality?

2. Line 165, " The proof can be found in, for example, Lemma 1 of [16]": It seems that the reference is not accurate (I did not find this result in the mentioned paper).

3. Lines 447-448: what is $A'$?

4. Line 480: what is the first case?

5. I believe authors should provide the missing proofs (I mentioned them in the Weaknesses section).

### Minor

1. Line 204: step sizes are $\eta_t$ are fixed $\to$ step sizes $\eta_t$ are fixed

2. Lemma 3.4: the summation is forgotten in the third term of the main inequality.

3. Line 430: should be Lemma 3.2 instead of Lemma 5

4. Line 439: the second inequality holds with probability strictly smaller than $1$

5. Lines 484-485, second and third rows: should be $\lambda$ instead of $\lambda_t$.

6. Line 487: Lemma 3.6 $\to$ Proposition 3.6

7. After line 488, definitions of $C_2$ and $C_3$: should be "$=$" instead of "$\leq$".

8. After line 493: after the first inequality the square is missing.

**Limitations:**

The authors address the limitations of their work in detail.

---

> ### Author Rebuttal · Authors · 2023-08-09
>
> We thank we reviewer for pointing out the significance of our contributions and for the detailed review.
>
> **Regarding the deterministic term**: The power of T in the term is $O\left(T^{\frac{1-2p}{3p-2}}\right)$. We can see that as $p\to1$ the rate becomes precisely $O\left(T^{-1}\right)$, which is the rate of gradient descent.
>
> **Regarding the comparison with Sadiev et al. (2023) in the nonconvex case**: In fact, Sadiev et al. (2023) achieve a rate with a significant suboptimal dependency on $T$, namely $O\left(T^{\frac{1-p}{p}}\right)$, while we achieve the optimal bound $O\left(T^{\frac{2-2p}{3p-2}}\right)$. This is a $poly\ T$ improvement for $p<2$, which vastly dominates the $\log T $ terms arising from the higher dependence on $\log(1/\delta)$. The reviewer is right that our dependence on $\log(1/\delta)$ is slightly higher in the very small $\delta$ regime and we will discuss this in the next revision.
>
> **For the unknown $T$**: Thank you for pointing out the gap between known $T$ and unknown $T$. We agree that removing the extra $\log T$ for unknown $T$ is an interesting question. Indeed, this remains open even under the more restrictive sub-gaussian noise assumption instead of heavy-tailed noise.
>
> **Regarding the incomplete proofs**: We thought some of these steps can be derived similarly to other proofs in our paper or cited ones, so we left some details out for the conciseness of the paper. We will add in more details in the revision of the paper.
>
> **Regarding the accelerated algorithm**: We will add this additional assumption in the main text.
>
> **Question**:
>
> 1. Line 39: Yes, we try to convey that one can obtain a probability bound from in-expectation bound using Markov inequality.
>
> 2. Line 165: Indeed, the reference is not correct. The correct reference is should be Beygelzimer et al. (2011), in the proof of Theorem 1 Eq. (1)-(3).
>
> 3. Lines 447-448: We define $A'$ in Line 441 as a proxy for $A$, using the variables $Z_{t}$.
>
> 4. Line 480: The first case here we meant the case for known $T$. We forgot to rewrite this part when moving some of the theorems and proofs to the appendix. The proof for Lemma B1 is complete.
>
> 5. We will add in the missing details for the incomplete proofs in the next revision.
>
> Finally, thank you for the attention to details and catching some typos. We will fix these in our revision.
>
> References:
>
> Beygelzimer, A., Langford, J., Li, L., Reyzin, L., & Schapire, R. (2011, June). Contextual bandit algorithms with supervised learning guarantees. In Proceedings of the Fourteenth International Conference on Artificial Intelligence and Statistics (pp. 19-26). JMLR Workshop and Conference Proceedings.

---

> > ### Comment · Reviewer_2mhk · 2023-08-14
> > **Thank you for the clarifications**
> >
> > I thank the authors for the clarifications: my comments are adequately addressed.
> >
> > **Deterministic term.** I am sorry for the confusion: when $p \to 1$ this term becomes $O(T^{-1})$. I wanted to say that when $p \in (1,2]$ this term is worse than $O(T^{-1})$, e.g., for $p=2$, it is $O(T^{-3/4})$. Therefore, when $\sigma$ is small, the result is not optimal.
> >
> > **Discussion of the logarithms.** In view of the above comment, when $\sigma$ is small the term $O(\gamma T^{\frac{1-2p}{3p-2}})$ can be the main one. In this case, as I explained in the review, the result from Sadiev et al. (2023) can be better for small $\delta$.
> >
> > In any case, the authors should discuss the dependence on the logarithmic factors more in the final version (as the authors promised).
> >
> > ---
> > I want to keep my score unchanged, assuming that the authors will make the promised modifications in the final version of the paper.

---

> > > ### Author Response · Authors · 2023-08-15
> > >
> > > We thank the reviewer for the insightful feedback. We will make sure to include these discussions in the revision.

---

### Official Review · Reviewer_d6sm · 2023-07-06

**Soundness:** 3 good
**Presentation:** 3 good
**Contribution:** 3 good
**Rating:** 5
**Confidence:** 3

**Summary:**

This paper considers the high probability guarantees for clipped stochastic gradient descent with heavy-tailed noise, in the setting of Zhang et al. (2020) where the unbiased gradient noise has finite p-th moments, where $p\in(1,2]$. The bound obtained in the paper is time-optimal and in the non-convex case, matches the lower bound in the literature. The analysis is based on the novel whitebox approach that analyzes the generating function of a well chosen martingale difference sequence to obtain tighter rates for stochastic gradient methods.

**Strengths:**

The bound obtained in the paper is time-optimal and in the non-convex case, matches the lower bound in the literature.

The analysis has technical novelty that involves analyzing the generating function of a well chosen martingale difference sequence to obtain tighter rates for stochastic gradient methods.

**Weaknesses:**

In terms of weakness, I think the first weakness is that some of the model assumptions seem to be very strong. For example, it is assumed that $\mathbb{E}[\Vert\hat{\nabla}f(x)-\nabla f(x)\Vert_{\ast}^{p}|x]\leq\sigma^{p}$. This assumption does not seem to hold in the simple setting of mini-batch. Is it possible to extend your analysis to allow the relaxation of this assumption to something like $\mathbb{E}[\Vert\hat{\nabla}f(x)-\nabla f(x)\Vert_{\ast}^{p}|x]\leq\sigma^{p}(1+\Vert x\Vert^{q})$ where $q$ can be related to $p$?

In your main result, i.e. Theorem 3.1., you mainly discuss the dependence on time $T$. What you obtained is $O(T^{\frac{2-2p}{3p-2}})$. However, there are other terms with explicit constants in your bound, and how does your bound depend on $p$? Does it have monotonic dependence on $p\in(1,2]$ or not? It seems to me that as $p\rightarrow 1$, the upper bound you have depends on whether $8\gamma/\sqrt{L\Delta_1}}$ and $\sigma$ are bigger than $1$ or smaller than $1$ and you will get very different results in these two cases. Is this something you expected and is there any intuition behind it?

**Questions:**

(1)	On page 3, you cited some works about high probability convergence for noises with bounded variance and heavy tails. Are there some works on unbounded variance that you can cite?
(2)	On page 6, line 234, can you provide a reference to Ville’s inequality?

**Limitations:**

I did not see such discussions.

---

> ### Author Rebuttal · Authors · 2023-08-09
>
> We thank the reviewer for the comments.
>
> **Regarding the assumption** $E [ ||\widehat{\nabla}f(x)-\nabla f(x)||\_*^p | x ]\le\sigma^{p}$:
>
> The reviewer is correct that in the small mini-batch setting, the variance may not be bounded. The assumption to the bounded $p$-th moments is a step towards relaxing the bounded variance assumption, and this is motivated by observation in practice (see Zhang et al. (2020) and our literature review). All prior works in the literature of heavy-tailed noise make this assumption. For example, this assumption is used in Gorbunov et al. (2020), for $p=2$ (the strongest case), or Zhang et al. (2020) and Sadiev et al. (2023) (the same assumption). Cutkosky & Mehta (2021) use this assumption in addition to another assumption that $E[||\widehat{\nabla}f(x)||\_*^p | x ]\le G^p$. We also think that relaxing this assumption, such as to $E[||\widehat{\nabla}f(x)-\nabla f(x)||\_*^p | x ]\le \sigma^{p}(1+||x||^q)$, is an important and interesting question to study in the future.
>
> **Regarding the dependency on $p$**:
>
> Generally $p$ is treated as a fixed problem parameter, and as the reviewer suggest, the dependency on p of the constants in front is not necessarily monotonic, but depends on the regime. What we care most about  in our paper is the exponent of $T$. This is monotonic in $p$.
>
> **Questions**:
>
> 1. Among the works we cited, Sadiev et al. [27] and Liu et al. [18] consider the unbounded variance case (ie, the assumption, $E [ ||\widehat{\nabla}f(x)-\nabla f(x)||\_*^p | x ]\le\sigma^{p}$ for $1<p\le2$).
>
> 2. Ville's inequality is a classic inequality (Ville, 1939). See, for example, the Wikipedia page for Ville's inequality and references therein.
>
> Reference:
>
> Ville, J. (1939), Etude Critique de la Notion de Collectif., Gauthier-Villars, Paris.

---

> > ### Author Response · Authors · 2023-08-20
> > **Did we address the reviewer's concerns?**
> >
> > We hope that we addressed all of the reviewer's concerns. We are happy to answer any questions but the reviewer-author discussion period ends soon. As discussed in our response, our assumptions are the least restrictive compared to all previous work in the same line of research. Could the reviewer please consider adjusting the score in light of this comparison?

---

### Official Review · Reviewer_QDgN · 2023-07-06

**Soundness:** 3 good
**Presentation:** 3 good
**Contribution:** 2 fair
**Rating:** 6
**Confidence:** 4

**Summary:**

This paper provides theoretical guarantees for clipped gradient methods in the presence of heavy-tailed gradient noise distributions, where the noise has a bounded $p$th moment for some $1 < p \leq 2$. The convergence guarantees are established with high probability. The authors' approach distinguishes itself from previous work by deviating from the classical techniques rely on using concentration inequalities coupled with an inductive union bound argument to control the iterates across all iterations. Instead, they bound the moment generating function of a well chosen supermartingale sequence which allows them to enhance the convergence guarantees for a wide range of clipped gradient based algorithm. The authors analyze Clipped-SGD algorithm in the smooth non convex setting, Clipped-SMD and Clipped-ASMD in the smooth convex setting. Specifically the rates they obtain are time-optimal and align with the latest lower-bounds, i.e., respectively $O(T^{\frac{2-2p}{3p-2}})$, $O(T^{\frac{1-p}{p}})$ and $O(T^{\frac{1-p}{p}}\sigma + T^{-2})$.

**Strengths:**

The paper have a well-written and well-structured presentation. The authors stresses the importance of their work by giving many intuitions on their new proving techniques and give comparisons with related works.  The authors provide guarantees and parameter values for the three algorithms studied in the paper, both for a known time horizon $T$, and an unknown time horizon. The theoretical analysis sounds clear and easy to follow backed with insightful intuitions. The "whitebox" technic based on a well chosen supermatingale exhibits potential reusability for addressing other problems.

**Weaknesses:**

The paper appears to be quite incremental and technical lacking of substantial novel contributions. The three algorithms studied in the paper are already well-established methods, and aside from the application of the "whitebox" analysis technique, the convergence analysis for the algorithms follows conventional approaches commonly employed for clipped-gradient-based methods. Furthermore, the analysis of Clipped-SMD heavily depends on the hypothesis that there exists a bound $\nabla_1$ on the gradient at the initial point i.e., $ \lVert \nabla f(x_1) \rVert_* $ $ \leq \nabla_1$. If the optimal point $x_*$ does not lie in the domain $ \lVert\nabla f(x_1) \rVert_*$  can be estimated w.h.p but requires the knowledge of $\sigma$ which is quite restrictive.

**Questions:**

Q1) Is it possible to extend the "whitebox" technic to the analysis of the Clipped-SGD with momentum in the non-convex setting?

Q2) How the studied algorithms compares practically with the parameters in the paper against existing state-of-the-art clipped gradient methods?

**Limitations:**

I think that the guarantees on the accelerated stochastic mirror descent algorithm should be added to the main body of the paper.

---

> ### Author Rebuttal · Authors · 2023-08-09
>
> We thank the reviewer for the comments and feedbacks.
>
> As pointed out by reviewer 2mhk, we believe that our paper makes “an important contribution to the stochastic optimization literature”. This includes:
>
> 1. Achieving the optimal rates in $T$ in all considered cases, closing the upperbound and lowerbound. Especially in the nonconvex case, we improve the existing bound by $poly\ T$ factor for $p<2$ (from $O\left(T^{\frac{1-p}{p}}\right)$ in Sadiev et al. to $O\left(T^{\frac{2-2p}{3p-2}}\right)$ in our case). Even if we are willing to suffer $\log T$ factors, previous techniques only allow us to obtain $O\left(T^{\frac{1-p}{p}}\right)$. Obtaining optimal bounds requires new insights and techniques.
>
> 2. Providing an analysis that is applicable generally for known and unknown time horizon. Analysis for the unknown time horizon case has not been achieved before to the best of our knowledge.
>
> 3. Showing an approach that works for the general domain (for SMD). Existing analysis only applies for compact or unconstrained problems.
>
> We would like to stress that the existing approach does not seem to be able to obtain these results.
>
> **Regarding the knowledge of $\sigma$**: We highlight that existing Clipped-SGD and Clipped-SMD algorithms generally require the knowledge of $\sigma$ even when the domain is unconstrained and the optimal point lies in the domain. We strictly improve this and only require \sigma when the optimal point does not lie in the domain.
>
> **Questions**:
>
> Q1) We believe that as long as a high probability analysis relies on a Freedman's type concentration inequality, it would be possible to apply the whitebox approach that to obtain tighter bounds. Clipped-SGD with momentum, such as in Cutkosky & Mehta (2021), can be proved using a Freedman's type concentration inequality, so we think it is possible to extend our techniques to this algorithm.
>
> Q2) Our new analysis of clipped gradient methods are mainly theoretically motivated, so we did not run any numerical analysis. However, we think the performance of the new parameter choices wouldn't differ too much from the state-of-the-art clipped algorithms, because the step sizes and clipping parameters are primarily obtained via hyperparameter tuning in practice.

---

> > ### Comment · Area_Chair_fpBC · 2023-08-16
> > **Please respond to the rebuttal, thanks!**
> >
> > Dear Reviewer QDgN,
> >
> > It would be nice if you could respond to the rebuttal.
> >
> > Thanks!
> >
> > AC

---

> > ### Comment · Reviewer_QDgN · 2023-08-17
> >
> > I would like to thank you for your response. I have no further concern.

---

> > > ### Author Response · Authors · 2023-08-20
> > >
> > > We thank the reviewer for the insightful feedback. Given that the reviewer has no further concerns, would it be possible for the reviewer to adjust the score accordingly?

---

### Official Review · Reviewer_Uh4e · 2023-07-07

**Soundness:** 4 excellent
**Presentation:** 3 good
**Contribution:** 3 good
**Rating:** 7
**Confidence:** 5

**Summary:**

In this paper, the authors consider clipped stochastic gradient methods: clipped-SMD in convex case (in non-convex case the y consider clipped-SGD), clipped-ASMD in convex case. They provided improved high-probability convergence analysis, which has some similar steps with analysis from (Gorbunov et.al 2020, Sadiev et.al 2023). Compare to this works, the authors provided analysis in non-Euclidean setup for clipped-SMD, clipped-ASMD. Also, they explain why the proposed approach works better than the previous one.



**Strengths:**

1. A good explanation of the difference between blackbox and whitebox approaches in a detailed way.
2. The authors provide the convergence analysis of the proposed methods in a non-euclidian setup.

Generally, the paper is well-written and has good results.

**Weaknesses:**

1. There are some issues dealing with clipped-ASMD. At the beginning of the analysis, they consider a more general assumption on the smoothness of $f$. But in the end, they use only standard smoothness assumption. Also, compare to clipped-SMD, they assume that $f(x^*) = 0$.
2. From my perspective, it would be better, if the proofs of all facts were provided in a more detailed way.

**Questions:**

1.  Could you explain why $z_k$ is decreasing sequence? It is not clear to me.
2.  About extension to non-smooth setting: How exactly could it be done? According to the proof and the final statement of the main fact about complexity, there is no constant $G$ in the final complexity. Could you comment on it?
3. The proof via blackbox approach is supported with a lack of explanation for some details, could you provide a full proof or the citation on the same proof technique? It is similar to the proof from the paper of Sadiev et.al, 2023.
4. Please, could you provide a small explanation for the third inequality in the last chain of inequality on page 19?
5. For inequality (d) on page 20, you did not write any explanation. Did you forget to do it?

Typos:
1. On line 204, the first 'are' is not needed.
2. In eq. (6), it will look better, If '(6)' is on the same row as expression.
3. On line 430, maybe, it would be better to write 'Lemma 3.2' or 'eq. (5)' instead of 'Lemma 5'
4. On lines 545 and 612, 'Proposition' is missed before '4.8'.



**Limitations:**

There are no limitations.

---

> ### Author Rebuttal · Authors · 2023-08-09
>
> We thank the reviewer for the compliments and feedback.
>
> **Regarding the weakness**:
>
> We use the general definition to address smoothness and non-smoothness in a unified way. The proofs for the non-smooth case follows very similarly to the smooth case. We give a sketch below in response to the reviewer's second question and will add this case to the revision of the paper.
>
> **We address the reviewer's questions below**:
>
> **Regarding the sequence $z_{t}$**:
> In Section 3, for example, we should have mentioned that we select the parameters $P_{t},Q_{t},\eta_{t},\lambda_{t}$ and ensure that $P_{t}\eta_{t}\lambda_{t}$ and $Q_{t}\eta_{t}^{2}\lambda_{t}^{2}$ are constants (Line 466). This guarantees that $z_{t}$ is a decreasing sequence. We will make this clear in the revision of the paper.
>
> **Regarding extension to non-smooth setting**: We can proceed similarly to the smooth case. Let $G$ be the Lipschitz constant. Starting with the basic inequality
>
> $\eta_{t}\Delta_{t} \le D_\psi(x^*,x_t)  -D\_\psi(x\^*,x\_{t+1}) $
>
> $\quad+\eta_t \langle x^*-x_t,\theta_{t}^{u} \rangle +\eta_t \langle x^*-x_t,\theta_{t}^{b} \rangle +\eta_{t}^{2}G^{2}$
>
> $\quad+2\eta_t^2 ( ||  \theta_{t}^{u}||\_{*}^{2} -E[|| \theta_{t}^{u}||\_*^2 | F_{t-1} ] )$
>
> $\quad+2\eta_t^2 E[|| \theta_{t}^{u} || \_{*}^{2} | F_{t-1} ]+2\eta_{t}^{2} || \theta_{t}^{b} ||\_*^2,$
>
> we define
>
> $Z_{t} =z_{t}\big(\eta_t \Delta_t -  D_\psi(x^*,x_t) + D\_\psi(x\^*,x\_{t+1})$
>
> $\quad- \eta_t \langle x^*-x_t,\theta_{t}^{b} \rangle -\eta_{t}^{2}G^{2}$
>
> $\quad - 2\eta_t^2 E[|| \theta_{t}^{u} || \_{*}^{2} | F_{t-1} ] -2\eta_{t}^{2} || \theta_{t}^{b} ||\_*^2 \big) $
>
> $\quad - (\frac{3}{8\lambda_{t}^{2}}+24z_{t}^{2}\eta_{t}^{4}\lambda_{t}^{2}) E[ ||  \theta_{t}^{u}||\_*^{2} | F_{t-1}] $
>
> where
>
> $z_{t}=\frac{1}{2P_{t}\eta_{t}\lambda_{t}\max_{i\le t}\sqrt{2D_\psi  (x^{*},x_{i})}+16Q_{t}\eta_{t}^{2}\lambda_{t}^{2}} $.
>
> For constants $c_{1}$ and $c_{2}$, in the case of known $T$, we choose
>
> $\lambda_{t}=\lambda=\max \\{ 2G, (\frac{26T}{\gamma} )^{1/p}\sigma \\} $
>
> $\eta_{t}=\eta=\min \\{ \frac{c_{2}}{G\sqrt{T}};\frac{c_{1}}{24\gamma\lambda} \\} =\min \\{ \frac{c_{2}}{G\sqrt{T}};\frac{c_{1}}{48G\gamma};\frac{c_{1}}{24\gamma}\left(\frac{26T}{\gamma}\right)^{-1/p}\sigma^{-1}\\} $
>
> and by similar analysis, we have
>
> $\frac{1}{T}\sum_{i=1}^{T}\Delta_{i} \le\frac{1}{2}(R_{1}+c_{1}+2c_{2})^{2}\max\\{ \frac{G}{c_{2}\sqrt{T}};\frac{48G\gamma}{c_{1}T};\frac{24\gamma^{\frac{p-1}{p}}\cdot26^{1/p}\sigma}{c_{1}}T^{\frac{1-p}{p}}\\}$.
>
> **Regarding the Blackbox proof**: The proof technique is generally similar to the approaches in Sadiev et al. (2023) and Gorbunov et al. (2020; for noise with bounded variance), so we only provide a sketch in our paper. One important note is that the existing bound in Sadiev et al. (2023) is suboptimal in $T$. In Remark 3.3, we explain the reason for this and how to make the analysis optimal. We will add in the complete analysis for the Blackbox case in our revision.
>
> **Regarding the inequality on page 19**: We use the inequality $ax-x^{2}\le\frac{1}{4}a^{2}$. The third inequality of the last chain on page 19 follows from combining two inequalities
>
> $\eta_{t} \langle \theta_{t},x_{t}-x_{t+1} \rangle -\frac{1}{4} ||  x_{t+1}-x_{t} ||^{2}  \le\eta_{t} ||  \theta_{t} ||\_{*} || x_{t}-x_{t+1} ||-\frac{1}{4} ||  x_{t+1}-x_{t} ||^{2} \le\eta_t^2 || \theta_{t} ||\_*^2$
>
> $G\eta_{t} ||  x_{t}-x_{t+1} || -\frac{1}{2} || x_{t+1}-x_{t} ||^{2}	\le 2G^{2}\eta_{t}^{2}.$
>
> **On Page 20**, we indeed forgot to explain (d), which follows from $|| \theta_{t}^{u}||\_{*}^{2}\le 4\lambda_{t}^{2}$ (Lemma 2.1, Eq (2)). We apply this on the term $E[ ||\theta_t^u ||_*^4 ] \le 4\lambda_t^2 E [||\theta_t^u||\_*^2]$ and obtain (d). We will add this explanation in our next revision.

---

> > ### Comment · Reviewer_Uh4e · 2023-08-13
> >
> > Thank you for your explanation, which allows me to understand your work better!

---

### Decision · Program_Chairs · 2023-09-21

**Decision:**

Accept (spotlight)

**Comment:**

All reviewers suggested acceptance and there is hence full consensus on the recommendation. I concur. There is no reason to be verbose in this metareview -- it is enough to refer to the original reviews. Congratulations on a nice paper.